# Repurposing the yellow fever vaccine for intratumoral immunotherapy

Maria Angela Aznar[1,*,†] , Carmen Molina[1], Alvaro Teijeira[1,2,3], Inmaculada Rodriguez[1,2,3], Arantza Azpilikueta[1,3], Saray Garasa[1,3], Alfonso R Sanchez-Paulete[1,‡], Luna Cordeiro[1,3], Iñaki Etxeberria[1], Maite Alvarez[1], Sergio Rius-Rocabert[4,5], Estanislao Nistal-Villan[4,5], Pedro Berraondo[1,2,3] & Ignacio Melero[1,2,3,**]

## Abstract

Live 17D is widely used as a prophylactic vaccine strain for yellow fever virus that induces potent neutralizing humoral and cellular immunity against the wild-type pathogen. 17D replicates and kills mouse and human tumor cell lines but not non-transformed human cells. Intratumoral injections with viable 17D markedly delay transplanted tumor progression in a CD8 T-cell-dependent manner. In mice bearing bilateral tumors in which only one is intratumorally injected, contralateral therapeutic effects are observed consistent with more prominent CD8 T-cell infiltrates and a treatment-related reduction of Tregs. Additive efficacy effects were observed upon co-treatment with intratumoral 17D and systemic anti-CD137 and anti-PD-1 immunostimulatory monoclonal antibodies. Importantly, when mice were preimmunized with 17D, intratumoral 17D treatment achieved better local and distant antitumor immunity. Such beneficial effects of prevaccination are in part explained by the potentiation of CD4 and CD8 T-cell infiltration in the treated tumor. The repurposed use of a GMP-grade vaccine to be given via the intratumoral route in prevaccinated patients constitutes a clinically feasible and safe immunotherapy approach.

**Keywords** 17D; cancer immunotherapy; intratumoral administration; virotherapy; yellow fever vaccine

**Subject Categories** Cancer; Immunology

See also: **TC Wirth et al** (January 2020)

## Introduction

Intratumoral administration of immunotherapy agents is a strategy that aims to achieve better efficacy while minimizing systemic toxicity (Aznar *et al*, 2017; Marabelle *et al*, 2018). Engineered viruses selectively replicating in tumors (virotherapy) constitute an area of active development in which the therapeutic agents can be given systemically or locally (Turnbull *et al*, 2015; Bommareddy *et al*, 2018; Pol *et al*, 2018). In recent years, evidence has been accumulating that the ensuing antitumor cytolytic immune responses constitute the main mechanism of action of virotherapy rather than direct cytopathic effects (Bommareddy *et al*, 2018). As a result, immune-potentiating transgenes are usually cloned in the vectors to enhance efficacy (Hu *et al*, 2006; Kim *et al*, 2006; Zhang *et al*, 2011; Goins *et al*, 2014; Quetglas *et al*, 2015).

Reminiscent of the intratumoral administration of live bacteria by William Cooley in the late 19[th] century (Coley, 1906, 1910), several groups have focused on the injection of replication-competent viruses into the tumor microenvironment of an injectable lesion seeking responses in distant metastases that are termed abscopal or anenestic effects (Aznar *et al*, 2017; Marabelle *et al*, 2018). In this regard, agents based on herpes simplex virus (HSV) (Goins *et al*, 2014), vaccinia virus (Kim *et al*, 2006), vesicular stomatitis virus (Patel *et al*, 2015; Kim *et al*, 2017), adenovirus (Jiang *et al*, 2017), new castle disease virus (NDV) (Lorence *et al*, 1994; Zamarin *et al*, 2014), and reovirus (Rajani *et al*, 2016; Samson *et al*, 2018) are the most advanced in their paths toward clinical use. The only FDA-approved agent to date is the HSV-based vector T-vec (Talimogene laherparepvec) because of its activity against unresectable or meta-static cutaneous melanoma (Andtbacka *et al*, 2015, 2016). In this case, the deletion-mutant attenuated virus is engineered to express GM-CSF and is used in patients preimmunized with the virus for safety reasons. Importantly, intratumoral T-vec seems to be

1   Center for Applied Medical Research (CIMA), University of Navarra, Pamplona, Spain
2   CIBERONC, Madrid, Spain
3   Instituto de investigación de Navarra (IDISNA), Pamplona, Spain
4   Microbiology Section, Dpto. CC. Farmaceuticas y de la Salud, Facultad de Farmacia, Universidad CEU San Pablo, CEU University, Boadilla del Monte, Madrid, Spain
5   Instituto de Medicina Molecular Aplicada (IMMA), Universidad CEU San Pablo, Pablo-CEU, CEU Universities, Boadilla del Monte, Madrid, Spain
    *Corresponding author. Tel: +34 948194700; Fax: +34 948194717; E-mail: maznargo@alumni.unav.es
    **Corresponding author. Tel: +1 2155 734187; Fax: +34 948194717; E-mail: imelero@unav.es
    †Present address: Center for Cellular Immunotherapies, Perelman School of Medicine, University of Pennsylvania, Philadelphia, PA, USA
    ‡Present address: Department of Genetics and Genomic Sciences, Icahn School of Medicine at Mount Sinai, New York, NY, USA

clinically synergistic with the anti-PD-1 monoclonal antibody (mAb) pembrolizumab (Ribas *et al*, 2017), although confirmation is pending in an ongoing phase III clinical trial (NCT02263508). Moreover, clinical activity seems to be also enhanced by combination with the anti-CTLA-4 mAb ipilimumab (Puzanov *et al*, 2016). Reportedly, preimmunization to HSV does not hamper local injection therapy and is performed on purpose in the clinical setting as a safety measure.

Evidence generated from intratumoral injection with wild-type NDV in mice indicates that most of the beneficial effects are immune-mediated by effector CD8 T cells and a synergistic combination with anti-CTLA-4 mAb was observed against tumor lesions not directly injected with virus (Zamarin *et al*, 2014). These results raise the question as to whether other viruses not affecting humans could also be used (Lizotte *et al*, 2015). However, veterinary or plant viruses are presumably not well adapted to human receptors and might be underperforming, even if they provide abundant nucleic acid with viral features acting as potent immune adjuvants on human pattern recognition receptors (PRRs). Interestingly in the case of NDV, it has been reported that preimmunization with virus not only does not hamper the therapeutic effects of local virotherapy but also actually enhances efficacy in mouse models (Ricca *et al*, 2018). Overall efficacy in virotherapy is probably the combined result of immunogenic cell death (Garg *et al*, 2017), ligands for pattern recognition receptors (Melero *et al*, 2015), and the ensuing immune response.

Yellow fever is a serious infectious condition caused by a prototypic member of the Flavivirus genus and transmitted by Aedes Aegypti mosquito bites (Monath & Vasconcelos, 2015). Attenuation of this RNA(+) virus by serial passage (18 times in mouse embryo tissues, 58 times in minced whole chick embryo tissue, and 128 times in minced chick embryo without nervous tissue) (Lloyd *et al*, 1936) led to the development of the Nobel-laureated vaccine (Lemmel, 2001). The vaccine is fully sequenced, shows a good safety profile in immunocompetent adults, and is used prophylactically to prevent the disease in endemic regions and in travelers (World Health, 2017). Attenuation is related to a few nucleotide differences with the Asibi genome (encoding for 31 amino acid substitutions) and to reduction in the quasispecies diversity in the viral population (Hahn *et al*, 1987; Beck *et al*, 2014).

Cancer treatment has been revolutionized by immunomodulatory monoclonal antibodies blocking the co-inhibitory receptor/ligand pair PD-1 and PD-L1 (Ribas & Wolchok, 2018). Moreover, strategies encompassing agonist monoclonal antibodies for costimulatory immune cell receptors such as CD137 (Morales-Kastresana *et al*, 2013) show potent effects against engrafted mouse tumors, are being clinically developed, and offer opportunities for synergistic combinations (Weigelin *et al*, 2015).

We reasoned that cancer virotherapy could benefit from repurposing the use of an already approved and widely used live attenuated viral vaccine. In that regard, 17D yellow fever vaccine was considered an advantageous alternative, since most humans in Western countries are naïve to the natural pathogen or to the vaccine. Vaccination induces very potent cellular and humoral immunity resulting in strong and long-lasting CD8 T-cell memory and potently induces type I IFN (Gaucher *et al*, 2008; Bassi *et al*, 2015; Fuertes Marraco *et al*, 2015). In this study, we show that the yellow fever vaccine can be safely injected intratumorally in mice,

giving rise to immune-mediated antitumor effects that can be combined with other immunotherapy agents in a clinically feasible fashion.

# Results

## 17D is cytopathogenic on an array of transplantable mouse and human cancer cell lines

Infectivity of mouse and human tumors is considered a prerequisite for antitumor activity. Hence, we explored whether the 17D yellow fever virus vaccine strain (Stamaril) could kill a panel of tumor cell lines. Figure 1A shows that all mouse cell lines tested are susceptible to 17D infection, even though with different multiplicity of injection (MOI) requirements. Furthermore, similar data were obtained with a panel of human tumor cell lines representing colon cancer, renal cell carcinoma, breast cancer, and melanoma (Fig 1B), while non-transformed human fibroblasts were resistant to such cytopathic effect at the same range of MOIs (Fig 1C). Therefore, the 17D live attenuated strain is able to effectively infect and induce cell death of mouse and human tumor cell lines of different tissue origin at virus concentrations that are innocuous for non-transformed human cells. Interestingly, all murine cell lines tested are amenable to engraftment in mice for *in vivo* experimentation.

## Intratumoral administration of 17D controls MC38 and B16OVA tumor progression

MC38 and B16OVA cancer cell lines were among those susceptible to 17D infection in culture (Fig 1A). We next examined the effect of 17D upon repeated intratumoral administration into MC38 (Fig 1D and E) or B16-OVA (Fig 1F and G) established subcutaneous tumors. Although treatment was not curative in any case, a clear delay in tumor progression was observed. To exert such an effect, the 17D virus had to be competent since UV-inactivated viral particles failed to control tumor growth (Fig EV1A and B) when similarly injected.

Having proven a reproducible control of tumor growth following intratumoral administration of 17D, we addressed the issue of whether a contralateral tumor (not directly injected) could be controlled as well in bilateral MC38 tumor models (Fig 1H–J). Experiments in Fig 1J show that certain contralateral therapeutic effects could be observed, although this statistical difference was lost at later time points (Fig 1I). Notably, such efficacy could not be attributed to the direct viral infection of the non-injected tumor, since 17D nucleic acid sequences were not detected in distant tumors of 17D-treated mice by a sensitive quantitative RT–PCR assay (Fig EV2A and B).

## 17D therapeutic effects are mediated by CD8 T cells

Next, we examined the necessary contribution of the different lymphocyte subsets by selective depletion of CD4 and CD8 T cells in bilateral MC38 tumor-bearing mice. Experiments shown in Fig 2A demonstrate that that CD8 T-cell depletion with an anti-CD8β mAb completely abolished 17D-induced tumor control in the directly injected (Fig 2B) and in the contralateral tumor nodules (Fig 2C),

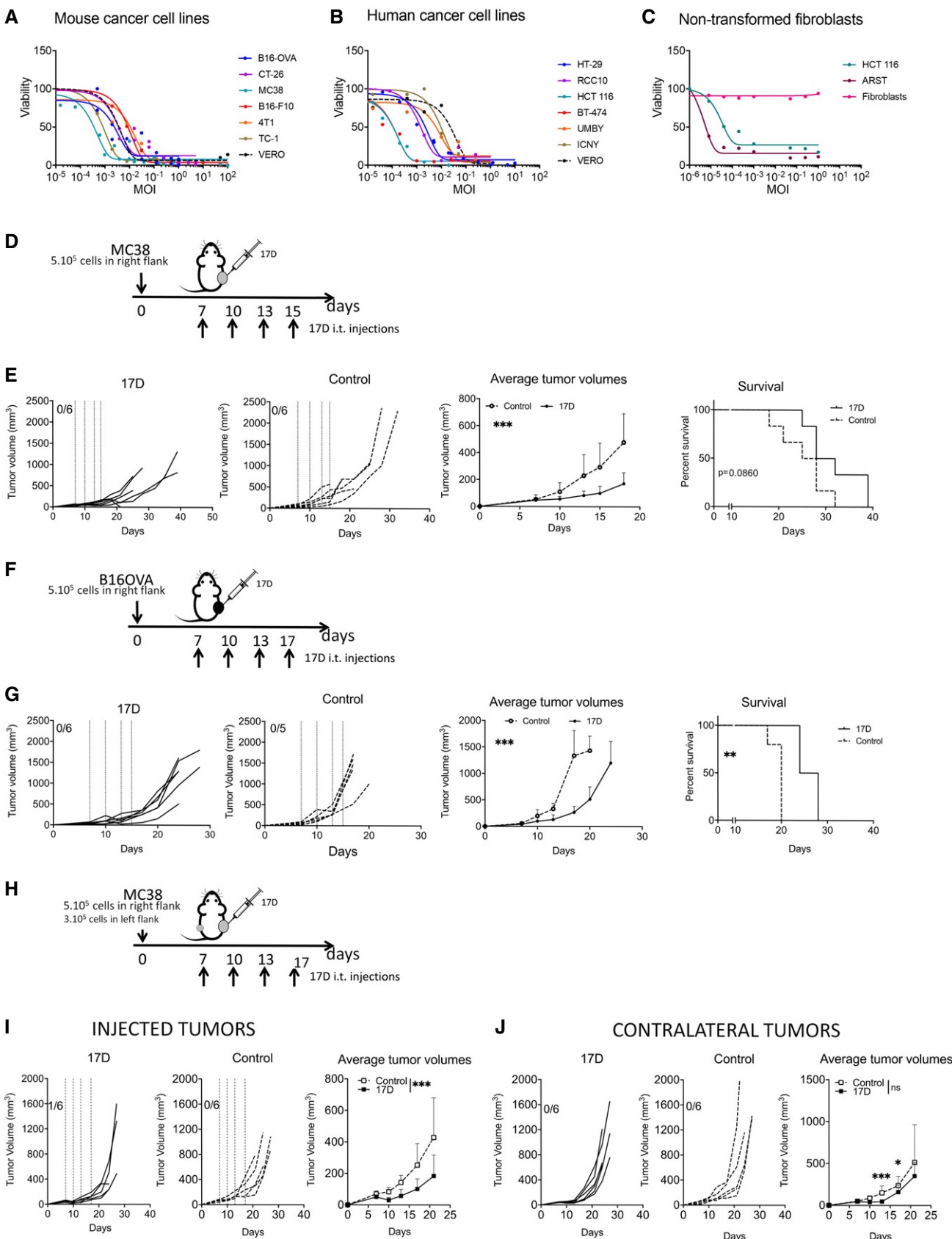

Figure 1.

**Figure 1. 17D antitumor effects *in vitro* and *in vivo*.**

A–C   The indicated mouse (*n* = 7) (A) and human (*n* = 7) (B) tumor cell lines, and non-transformed human fibroblasts (C) were exposed to increasing MOIs of 17D virus in culture, and subsequently cell viability was assessed by crystal violet staining performed 6 days later. The Vero cell line used for viral production is included as a positive control in each experiment with tumor cell lines, and the human tumor cell lines ARST1 and HCT 116 were included as positive controls of infection for the experiments with human non-transformed fibroblasts. Results shown are representative of at least three experiments similarly performed.

D, E   MC38 tumors were engrafted and treated as schematically represented. Graphs represent individual tumor size follow-up upon intratumoral injections with 17D (*n* = 6) or vehicle (*n* = 6) as a control that are also shown as mean ± SD and as overall survival of the mice. Dashed lines indicate the injection days of 17D. ***P < 0.001.

F, G   B16OVA-derived melanomas were engrafted and treated as schematically represented. Graphs represent individual tumor size follow-up upon intratumoral injections with 17D or vehicle as a control (*n* = 6 and *n* = 5 for 17D and control groups, respectively), also shown as mean ± SD and as overall survival of the mice. Dashed lines indicate the injection days of 17D. ***P < 0.001, **P < 0.01.

H   Schematic representation of the experiments to engraft and treat MC38-derived bilateral subcutaneous tumors.

I   Individual tumor growth follow-up and mean ± SD of tumor lesions directly injected with 17D or control vehicle (*n* = 6 per group). Dashed lines indicate the injection days in injected tumors. ***P < 0.001.

J   Follow-up of contralateral non-injected tumors and mean ± SD of tumor lesions in the same mice as in (I). ***P < 0.001, *P < 0.05.

Data information: Mean tumor volume growth over time was fitted using non-linear regression curve fit. Treatments were compared using the extra sum-of-squares *F*-test. Mantel–Cox test was used for survival analysis. Experiments are representative of at least two similarly performed. ***P < 0.001, **P < 0.01.

Source data are available online for this figure.

and lead to reduced overall survival (Fig 2D). In the case of CD4 depletion, no effect was seen in the treated tumor, but the absence of CD4 T cells improved the therapeutic effects in the contralateral tumor. Experiments in Fig EV3A–C depleting granulocytes and myeloid suppressor cells or NK/NKT cells yielded negative results, indicating that these leukocyte subsets do not have a necessary role in the therapeutic effects of 17D intratumoral injections. Selective leukocyte depletions were confirmed in every experiment in peripheral blood (Fig EV3D).

## Intratumoral 17D used in combination with immunomodulatory anti-CD137 and anti-PD-1 monoclonal antibodies

Given the partial efficacy of the intratumoral 17D-mediated by CD8 T-cell responses, we investigated if this treatment strategy could be enhanced by mAbs acting on costimulatory or co-inhibitory pathways (Melero *et al*, 2007).

Setting up bilateral MC38-derived tumor nodules, we tested combinations of repeated intratumoral 17D and systemic anti-CD137 or anti-PD-1 mAbs (Fig 3A). Very clear synergistic tumor-eradicating responses were observed in directly injected tumors when 17D is combined with systemic anti-CD137 mAb (Melero *et al*, 1997) that was almost ineffective by itself (Fig 3B). Anti-PD-1 mAb, also ineffective by itself, showed mild local additive efficacy when used in combination with intratumoral 17D regimen (Fig 3B). Regarding contralateral efficacy, a certain delay in tumor growth was achieved in mice contralaterally treated with 17D and systemic anti-PD-1 mAb. Systemic anti-CD137 by itself delayed contralateral tumor growth to a similar degree without further improvement by 17D treatment of the contralateral tumor (Fig 3C). This leads to an improved overall survival of the mouse group treated with 17D + anti-CD137 (Fig 3D). These observations, especially the efficacy achieved in 17D-injected tumors, suggest the clinical interest of anti-CD137-targeted agonists (Compte *et al*, 2018) to be used in conjunction with intratumoral 17D.

## Immune events underlying intratumoral 17D therapeutic effects

CD8-mediated antitumor responses are completely dependent on conventional type I dendritic cells (cDC1) (Hildner *et al*, 2008;

Murphy *et al*, 2016; Sanchez-Paulete *et al*, 2017). Such antigen-presenting cells that mediate CD8 T-cell crosspriming (Sanchez-Paulete *et al*, 2017) of tumor antigens are defective in BATF3$^{-/-}$ mice showing that they are required for a number of successful immunotherapies to work (Salmon *et al*, 2016; Sanchez-Paulete *et al*, 2016; Spranger *et al*, 2017). Consistent with this notion, the tumor growth delay mediated by intratumoral 17D injection was lost in BATF3$^{-/-}$ mice (Fig 4A–C), especially in non-injected contralateral tumors (Fig 4C).

We have previously reported that antitumor immunotherapy elicited by anti-CD137 and anti-PD-1 mAbs was totally dependent on the performance of cDC1 cells in tumor antigen crosspriming (Sanchez-Paulete *et al*, 2016). In keeping with these findings, the local synergistic combinations of 17D and systemic anti-CD137 mAb also lost their efficacy (Fig 4B and C).

Having determined that 17D efficacy is mediated by CD8 T-cell responses, 17D intratumoral injections were predicted to augment immune cell infiltrates and induce parallel changes in tumor-draining lymph nodes (TDLN) toward which the virus and antigen-presenting cells ought to be drained. In this regard, as early as 5 days following 17D intratumoral treatment regimens (Fig 5A), the absolute number of CD8 T cells per gram of tumor was increased (Fig 5B), while CD4$^+$FOXP3$^+$ Tregs were markedly reduced in their numbers (Fig 5C), resulting in high CD8/Treg and conventional CD4 T (Tconv)/Treg ratios (Fig 5D). In this context, the number of infiltrating NK1.1$^+$ cells did not increase (Fig 5E). In Fig 5F, we analyzed whether 17D treatment modified the expression of lymphocyte-targeted receptors for immunostimulatory mAbs, observing an increase in CTLA-4 surface expression on CD4 and CD8 T cells. In contrast, CD137 and PD-1 expression did not increase, likely indicating a less exhausted T-cell phenotype (Apetoh *et al*, 2015; Williams *et al*, 2017).

17D intratumoral injection also induced an increase in CD8 T cell in non-injected tumors, and in these locations, more cells expressed CD137 and/or PD-1 as compared to controls (Fig 5G and H). In these contralateral lesions, NK cell numbers increased and upregulated CD137 on their surface as an activation marker (Fig 5I and J).

Tumor-draining lymph nodes from intratumorally 17D-treated mice were larger, reflecting an increase in the content of CD8,

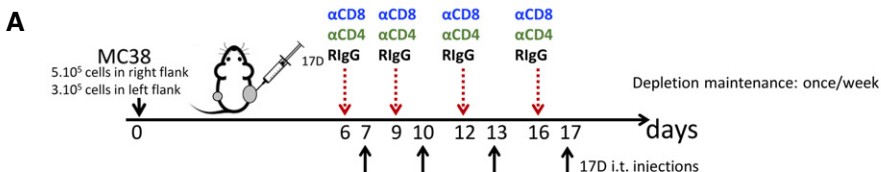

# Figure 2. T-cell requirement for the antitumor effects of 17D intratumoral injections.

A  Schematic representation of the experiments upon treatment of mice bearing bilateral MC-38-derived colon carcinomas when concurrently eliminating CD4 or CD8 T cells with depleting monoclonal antibodies as indicated.

B  Individual follow-up and mean ± SD ($n$ = 6 per group) of tumor sizes in the directly 17D or control vehicle-injected tumors. Dashed lines indicate the days of intratumoral injection.

C  Individual tumor growth and mean ± SD follow-up of the size of contralateral (non-directly injected) tumors under the same indicated conditions ($n$ = 6 per group).

D  Overall survival of the indicated groups of mice ($n$ = 6 per group).

Data information: Mean tumor volume growth over time was fitted using non-linear regression curve fit. Treatments were compared using the extra sum-of-squares $F$-test. Mantel–Cox test was used for survival analysis. ***$P$ < 0.001, **$P$ < 0.01, *$P$ < 0.05, ns: non-significant.

conventional CD4 cells, NK cells, and to a lesser extent, Tregs (Fig EV4A), resulting in an increase in the Tconv/Treg ratio (Fig EV4B). In these organs, more CD8 and CD4 T cells expressed CD137 and PD-1 on their surface upon 17D intratumoral treatments in a way that such cells would become susceptible to stimulation by the corresponding immunomodulatory mAbs (Fig EV4C).

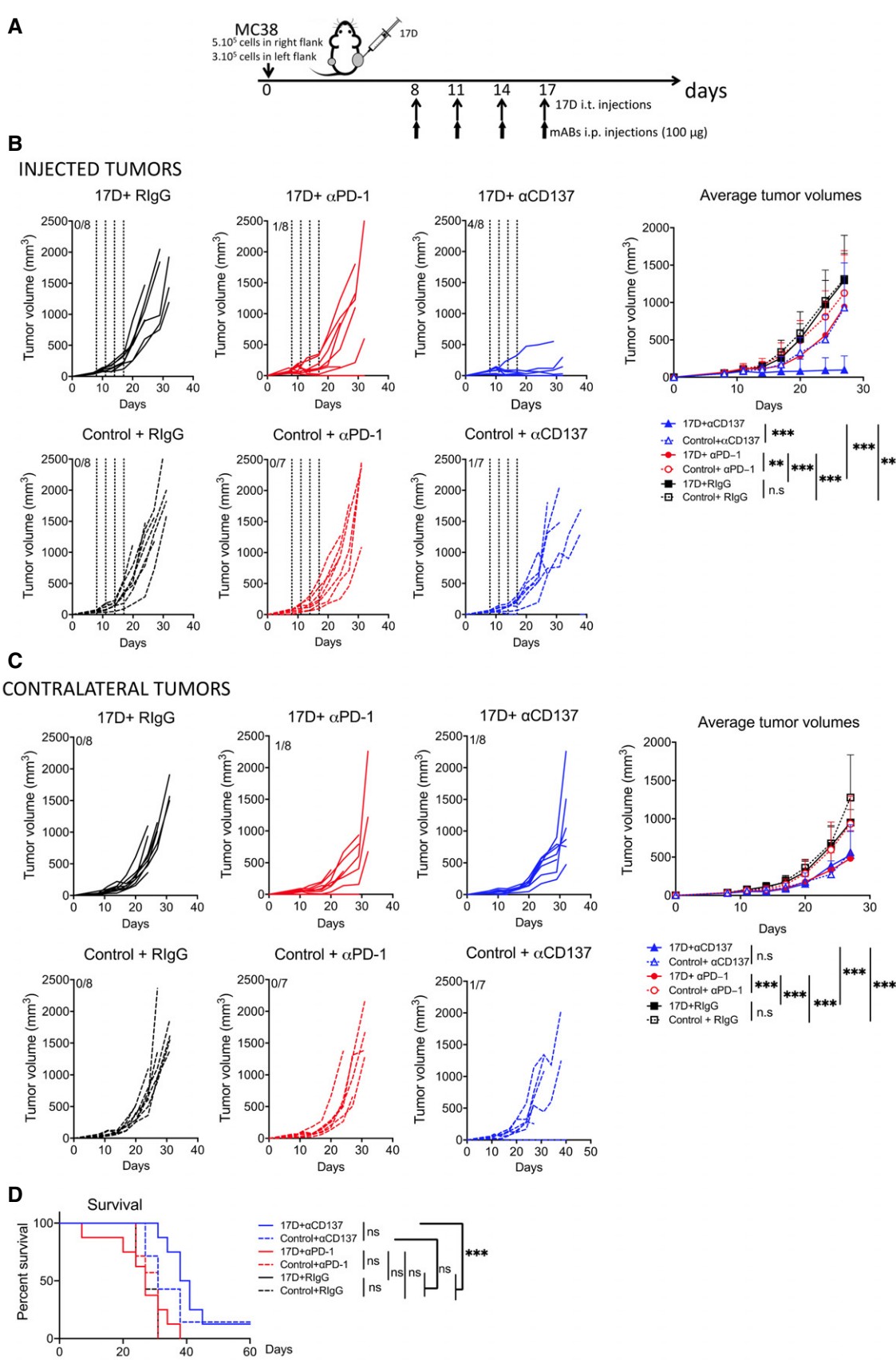

**Figure 3.**

**Figure 3.  Combinations of intratumoral 17D virotherapy and systemic administration of anti-PD-1 and anti-CD137 immunomodulatory monoclonal antibodies.**

A   Schematic representation of the experiments performed in mice bearing bilaterally MC38-derived tumors treated on the indicated days with intratumoral 17D virus or control vehicle in one of the lesions and intraperitoneally with the indicated immunotherapeutic monoclonal antibodies anti-PD-1, anti-CD137, or control RIgG.
B   Individual and averaged (mean ± SD) tumor size progression of the indicated experimental groups. Dashed lines indicate the days of intratumoral injection.
C   Individual and averaged (mean ± SD) tumor size follow-up of the contralateral tumors that were not intratumorally injected.
D   Overall survival of the indicated groups of mice.

Data information: *n* = 8 mice/group in all the groups except for Control + anti-PD-1 and Control + anti-CD137 (*n* = 7 mice/group). Mean tumor volume growth over time was fitted using non-linear regression curve fit. Treatments were compared using the extra sum-of-squares *F*-test.  Mantel–Cox test was used for survival analysis. Data are representative of two experiments identically performed. ***$P < 0.001$, **$P < 0.01$, *$P < 0.05$, ns: non-significant.

### 17D preimmunization enhances the efficacy of 17D intratumoral therapy

Preimmunization to 17D could be a safety feature but could also be deleterious for therapy, if neutralizing antibodies impede tumor cell infection. To evaluate the effects of 17D pre-existing immunity on the efficacy of intratumoral 17D, we immunized C57BL/6 mice 2 weeks before tumor cell inoculation and 21 days prior to 17D intratumoral therapy (Fig 6A). Such mice showed high serum titers of neutralizing antibodies that hamper the infection of Vero cell cultures by 17D (Fig 6B). In this setting, efficacy was not only preserved but also clearly increased, with an important further delay in tumor growth (Fig 6C and D).

Given this counterintuitive effect, we explored if T lymphocytes from immunized mice could transfer the favorable therapeutic effect. The experiments, performed as described in Fig 7A, showed that CD8 T cells transferred from immunized mice transmitted the survival benefit and the enhanced efficacy to intratumoral 17D in treated and distant tumor nodules (Fig 7B–D). This occurred when passive transfer with pre-immune CD8 cells occurred 24 h prior to 17D intratumoral treatment. In contrast, transfer of CD4 T cells or 0.25 ml of immune serum did not give rise to the additional benefit.

Five days after B16OVA bilaterally tumor-bearing mice have received a two-dose regimen of 17D, we explored T-cell infiltrates in mice preimmunized or not with the yellow fever vaccine (Fig EV5A). The most striking difference observed was a strong increase in the infiltration of CD4 T cells into the tumor microenvironment of preimmunized mice (Fig EV5B) that was observed in the treated tumors but not as clearly in the contralateral ones in which only a tendency was observed. These results on CD4 T-cell infiltrates are reminiscent of a previous report on intratumoral treatments with new castle disease virus (NDV) to preimmunized mice (Ricca *et al*, 2018). To analyze the CD8 compartment, we had adoptively transferred congenic OT-1 CD45.1 CD8 cells to these B16OVA tumor-bearing mice to be followed in addition to the endogenous CD45.2 endogenous CD8 T cells (Fig EV5A). At the time point chosen for analysis, a tendency to increase the content endogenous CD45.2 CD8$^+$ T cells was detected in treated and contralateral tumors of preimmunized mice (Fig EV5C and D). However, no increases at this time point were seen for tumor-specific OT-1 transferred cells in either the treated or non-treated site. Curiously, the PD-1-expressing endogenous CD8$^+$ T-cell subset was significantly enriched at this time point, although the levels of expression assessed by mean fluorescence intensity in this case were lower in comparison with those in non-immunized animals (Fig EV5C and D, upper panels). Adoptively transferred naïve OVA-specific CD45.1 OT-1 T cells present in 17D-treated and contralateral tumor nodules

were found at identical frequencies among the different experimental groups, showing again similar changes in terms of PD-1 expression levels in the preimmunized versus non-immunized cases (Fig EV5B, lower panels).

Taken together, these experiments indicate that recalled anti-17D CD4 and CD8 cellular immunity helped the onset or the amplitude of antitumor immunotherapeutic reactions.

### Intratumoral injection of 17D at earlier stages of tumor development exerts more pronounced antitumor efficacy

Although 17D exerted antitumor efficacy in MC38 and B16OVA preclinical models upon four intratumoral injections given since day 8, for a practical application purposes, we decided to test two intratumoral doses of virus but given to mice carrying MC38 tumors for only 6 days rather than the usual 8 days as in the previous experiments (Fig 8A). Of note, tumors at day 6 are already established and intratumoral injection of 17D is feasible. In this experimental setting, we observed a very marked antitumor response with only the two intratumoral 17D treatments (Fig 8B).

This level of efficacy on day 6 MC38 tumors was also observed when using 17D virus expanded in embryonated eggs as compared to 17D virus produced in Vero cell cultures (Fig 8C). This is considered important since the commercially available vaccine is produced using such methodology.

### Efficacy of 17D intratumoral therapy is dependent on type I IFN

We finally studied if antitumor efficacy was related to the type I interferon system as induced by the vaccine strain upon intratumoral injection. In MC38 tumors treated on days 6 and 8, immediate pretreatment with an anti-IFNα receptor neutralizing antibody abrogated the beneficial effect (Fig 8D). This indicates that IFNα/β induction and the function of these cytokines are involved in the train of events leading to tumor control.

## Discussion

In this study, we provide evidence of the potential of intratumoral yellow fever vaccine 17D for cancer immunotherapy. Our results show that such treatment enhances CD8 T-cell-mediated anti-cancer immunity.

The fact that 17D is a widely used prophylactic agent used for subcutaneous or intramuscular vaccination greatly simplifies eventual clinical development of the immunotherapy strategies incorporating 17D intratumoral injections (http://products.sanofi.com.au/

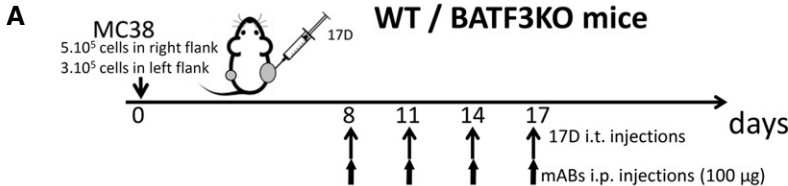

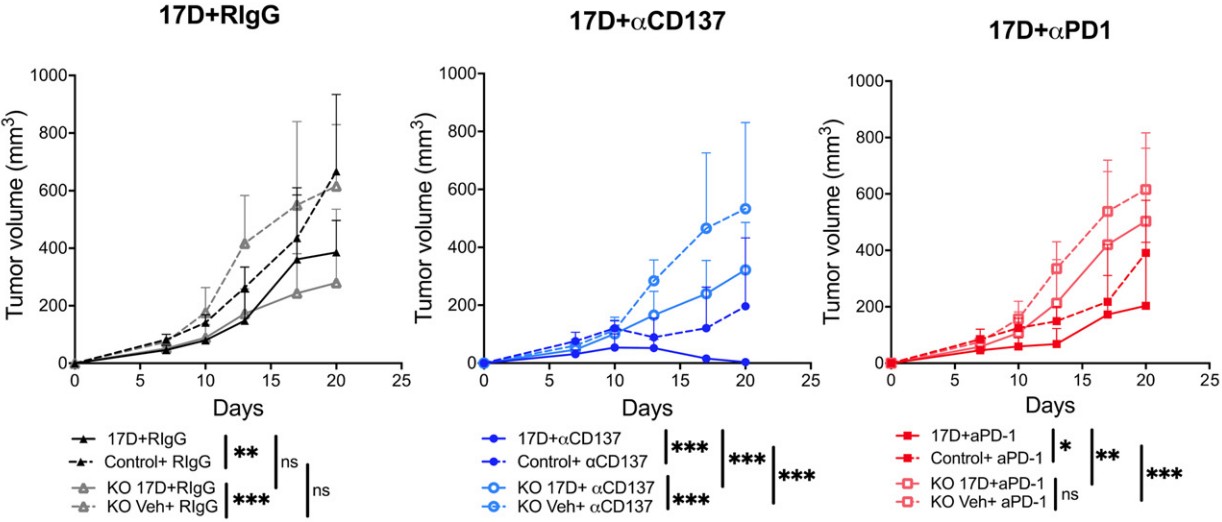

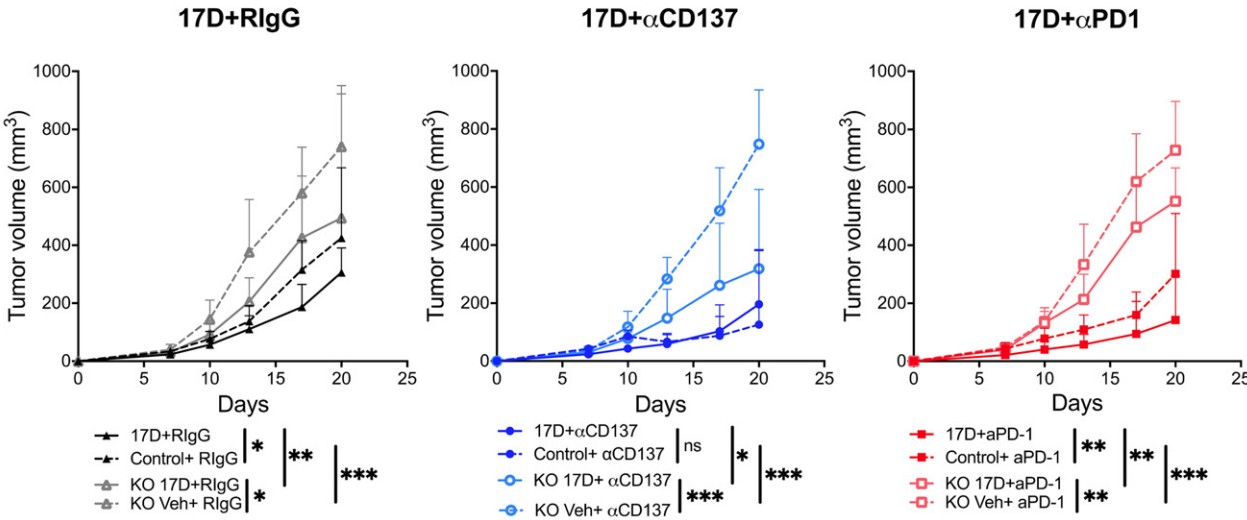

**Figure 4. Treatment efficacy reduction in BATF-3 knockout mice.**

A Scheme representing experiments as in Fig 2 performed in WT and BATF-3$^{-/-}$ mice bearing bilateral MC38 tumors and treated as indicated.

B, C Size follow-up of treated and contralateral tumors of the indicated groups of mice (mean ± SD), ($n$ = 5 for all 17D-injected groups, WT Control + RIgG $n$ = 6, WT Control + anti-PD-1 and WT Control + anti-CD137 $n$ = 7, KO Control + RIgG and KO Control + anti-CD137 $n$ = 6, and KO Control + anti-PD-1 $n$ = 7). Mean tumor volume growth differences were calculated with non-linear regression curve fit. Treatments were compared using the extra sum-of-squares $F$-test. ***$P$ < 0.001, **$P$ < 0.01, *$P$ < 0.05, ns: non-significant.

Source data are available online for this figure.

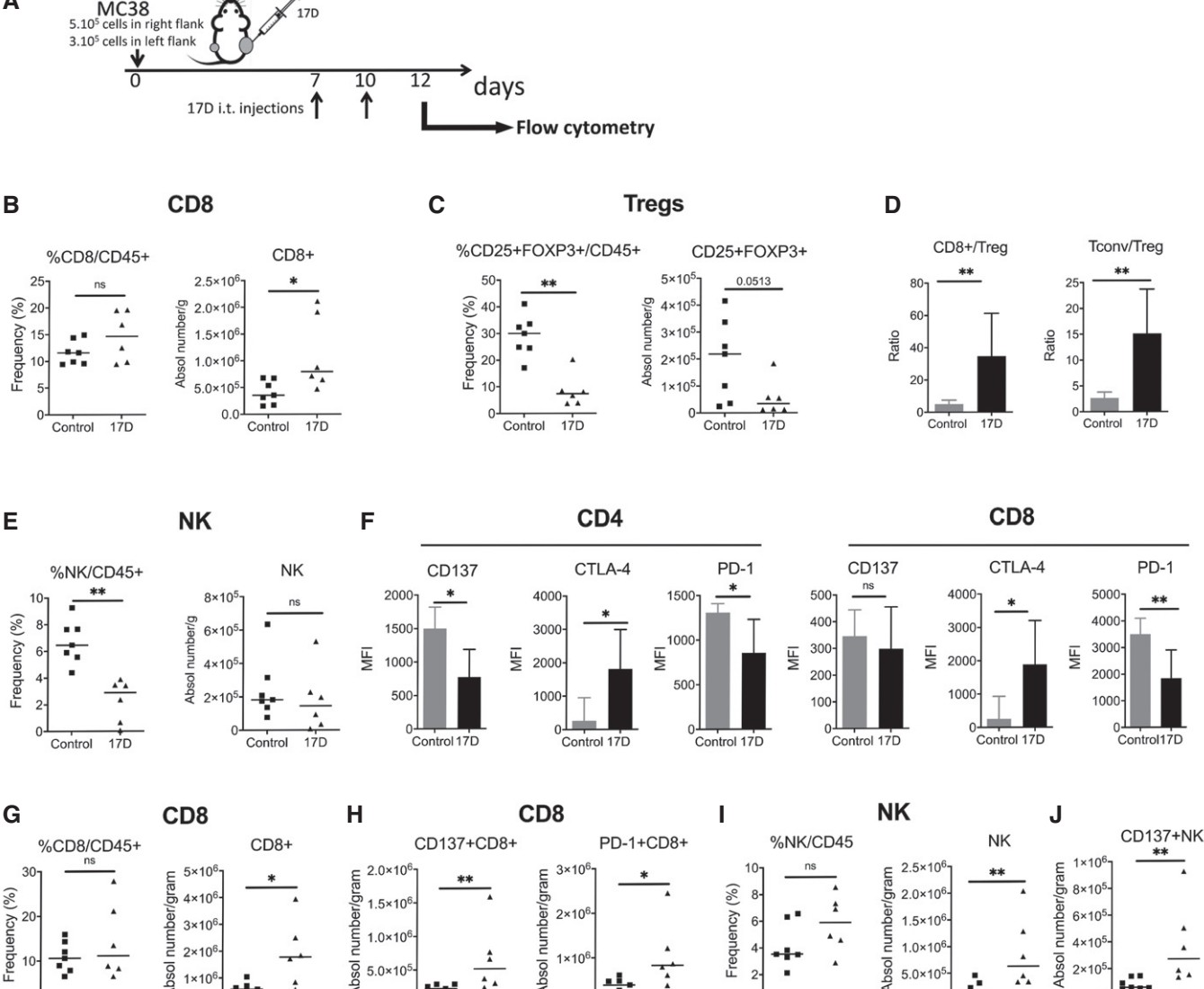

**Figure 5. Immune tumor infiltrates following 17D treatment.**

A Scheme representing one side only intratumoral treatment with 17D (*n* = 6) or control vehicle (*n* = 7) of mice carrying bilateral MC38-derived tumors. On day +12, single-cell suspensions were derived from the excised treated and contralateral tumor nodules. Cell suspensions were processed for multicolor flow cytometry assessments.

B–E FACS quantification of the content of the indicated CD8 (B), Treg (C), CD8- and CD4/Treg ratios (D), and NK lymphocyte subsets (E) in the directly treated tumors.

F Surface expression of CD137, CTLA-4, and PD-1 on gated CD4 and CD8 T cells in the same cell suspensions derived from directly treated tumors.

G Content of CD8 T cells in the contralateral non-directly treated tumors.

H Presence of CD8 subsets with surface expression of PD-1 or CD137.

I, J NK cell presence in contralateral tumors (I) showing the number of NK cells expressing surface CD137 in (J).

Data information: Data are shown as individual mice or bars depicting median ± SD. Two-tailed Mann–Whitney test was used to evaluate the statistical differences between both groups (*n* = 6 for 17D and *n* = 7 for control vehicle) and representing results from two experiments similarly performed. \**P* < 0.05, \*\**P* < 0.01, ns: non-significant.

---

vaccines/STAMARIL_NZ_CMI.pdf). In this regard, the fact that preimmunization does not hamper efficacy but, on the contrary, markedly enhances it provides an exciting safety feature that would allow higher doses of 17D to be used.

Although 17D is highly immunogenic, it remains to be seen if engineered versions that confer expression of immune-promoting

transgenes would be therapeutically more successful. Previous attempts to engineer 17D-based viral vectors have been made in the past with model tumor antigens (McAllister *et al*, 2000) with enhancing effects of the resulting constructs when administered subcutaneously or intravenously. However, it must be considered that the use of unmodified 17D (as such) has an extensive positive safety record

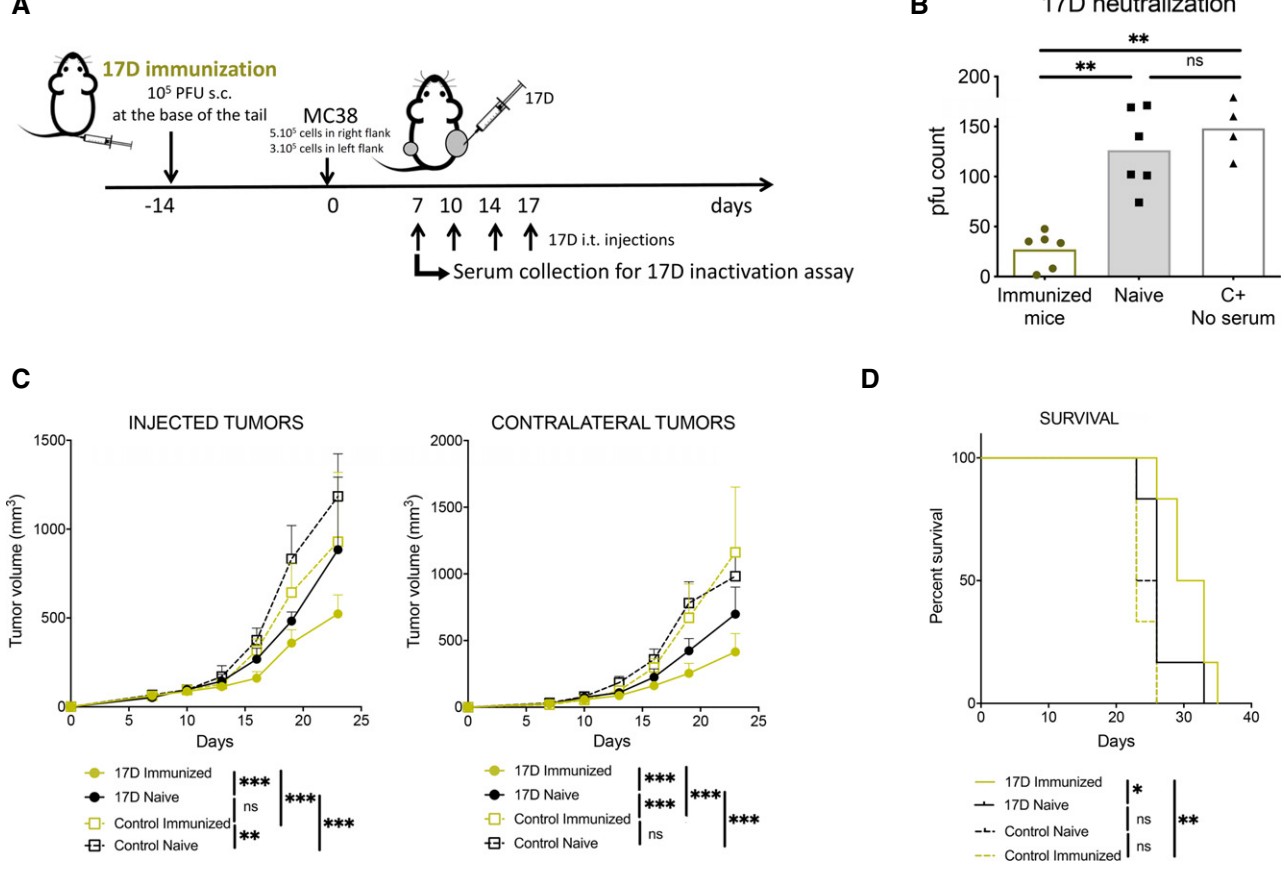

**Figure 6.  Pre-existing immunity to 17D virus improves the therapeutic effects of intratumoral 17D injections.**

A       Scheme representing the experiments in which mice were subcutaneously exposed to a subcutaneous vaccinating dose of 17D virus or control vehicle. On day 0, mice were bilaterally engrafted with MC38-derived tumors and treated when indicated with intratumoral doses of 17D or control vehicle only given to one of the lesions.

B       Neutralizing antibodies in the serum of naïve and immunized mice ($n = 6$/group) tested in terms of 17D virus infectivity on Vero cells in culture that was quantified as plaque-forming units on day 6 of culture (C+ samples are dilutions of 17D viral stocks, $n = 4$). Sera samples were drawn on day +7 following tumor cell inoculations. Kruskal–Wallis test  and Dunn's with multiple comparisons correction.

C, D    Tumor volume (mean ± SD) follow-up (C) and overall survival (D) of the indicated groups of mice ($n = 6$/group). Mean tumor volume growth differences were calculated with non-linear regression curve fit and treatments were compared using the extra sum-of-squares $F$-test. Mantel–Cox test was used for survival analysis.

Data information: Data shown are from a single experiment and represent results from three experiments similarly performed. *$P < 0.05$, **$P < 0.01$, ***$P < 0.001$, ns: non-significant.

Source data are available online for this figure.

(Thomas *et al*, 2012) and has been the object of much clinical experience, which when repurposing the vaccine for cancer immunotherapy greatly simplifies regulatory and pragmatic issues.

The mode of action upon local intratumoral release of 17D likely involves a train of events that entail tumor cell infection, tumor antigen crosspriming, and the infiltration of CD8 T cells into the tumor microenvironment. Contribution of viral RNA acting on TLR3, TLR7/8, or on the intracellular helicases (RIG-I and MDA5) is likely inducing maturation of DC for CD8 T-cell priming and local type I IFN release as previously shown for other viruses (Huarte *et al*, 2006; Melero *et al*, 2015). Once induced or reinvigorated, T-cell immunity can reach non-injected lesions reminiscent of what has been reported with T-vec in humans and with NDV in mice by Dr. Wolchok and Zamarin's group (Ricca *et al*, 2018). In keeping with these reports, we find clear increases in CD8 T cells infiltrated

injected tumors and enhanced active phenotype of CD8 tumor lymphocyte infiltrates in treated and distant lesions.

Our findings in preimmunized mice showing better antitumor activity are quite interesting since this will also mean an additional safety feature for translation to patients. Understanding the exact mechanism underpinnings of enhanced efficacy due to preimmunization could be complex. On the one hand, adoptive transfer of CD8 T cells from preimmunized mice seems to convey the beneficial effects, while on the other hand, the most striking difference in the tumors treated in preimmunized mice is a more prominent CD4 T-cell infiltrate, at least at day 5 on-treatment. The latter finding was also previously reported for NDV intratumoral treatments (Ricca *et al*, 2018). The exact role of infiltrating CD4 remains to be seen, and we are currently exploring the roles of CD40L and IFNγ expressed by antiviral CD4 cells as candidate mechanisms to explain

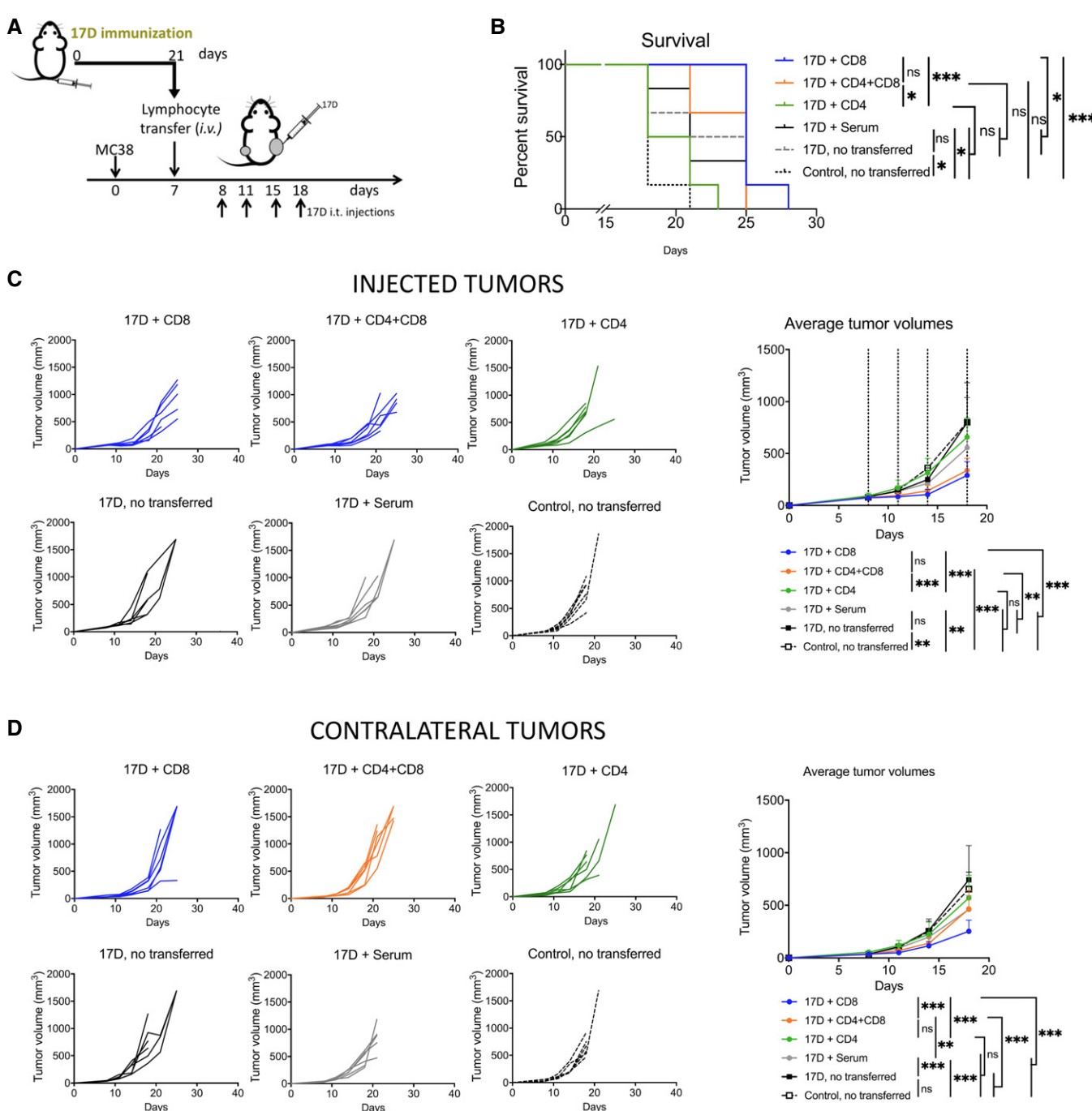

**Figure 7. Enhanced antitumor effects due to preimmunization are transferred along with CD8 T cells.**

A   Schematic representation of experiments in which 17D preimmunized donor mice were sacrificed to obtain CD8 and CD4 splenocytes and serum 21 days following subcutaneous vaccination. Mice bearing bilateral MC38 tumors were then intravenously transferred on day +7 with $10^7$ CD4 or/and CD8 T cells or 250 μl of serum.

B   Overall survival.

C   Individual and averaged ± SD follow-up of the tumor directly treated with 17D in the recipient mice. Dashed lines indicate the days of intratumoral injection.

D   Individual and averaged ± SD follow-up in the corresponding contralateral tumors.

Data information: Mean tumor volume growth over time was fitted using non-linear regression curve fit. Treatments were compared using the extra sum-of-squares *F*-test. Mantel–Cox test was used for survival analysis. Results are from a single experiment performed with *n* = 6 mice per group. *$P < 0.05$, **$P < 0.01$, ***$P < 0.001$.

increased efficacy of 17D and other viral vectors. Differences in CD8 T cells were more subtle and probably involve dimmer PD-1 expression levels in preimmunized cases denoting activation (Araki *et al*, 2013; Ahn *et al*, 2018) and not the higher levels seen on terminally exhausted lymphocytes (Blackburn *et al*, 2008; Wherry & Kurachi, 2015).

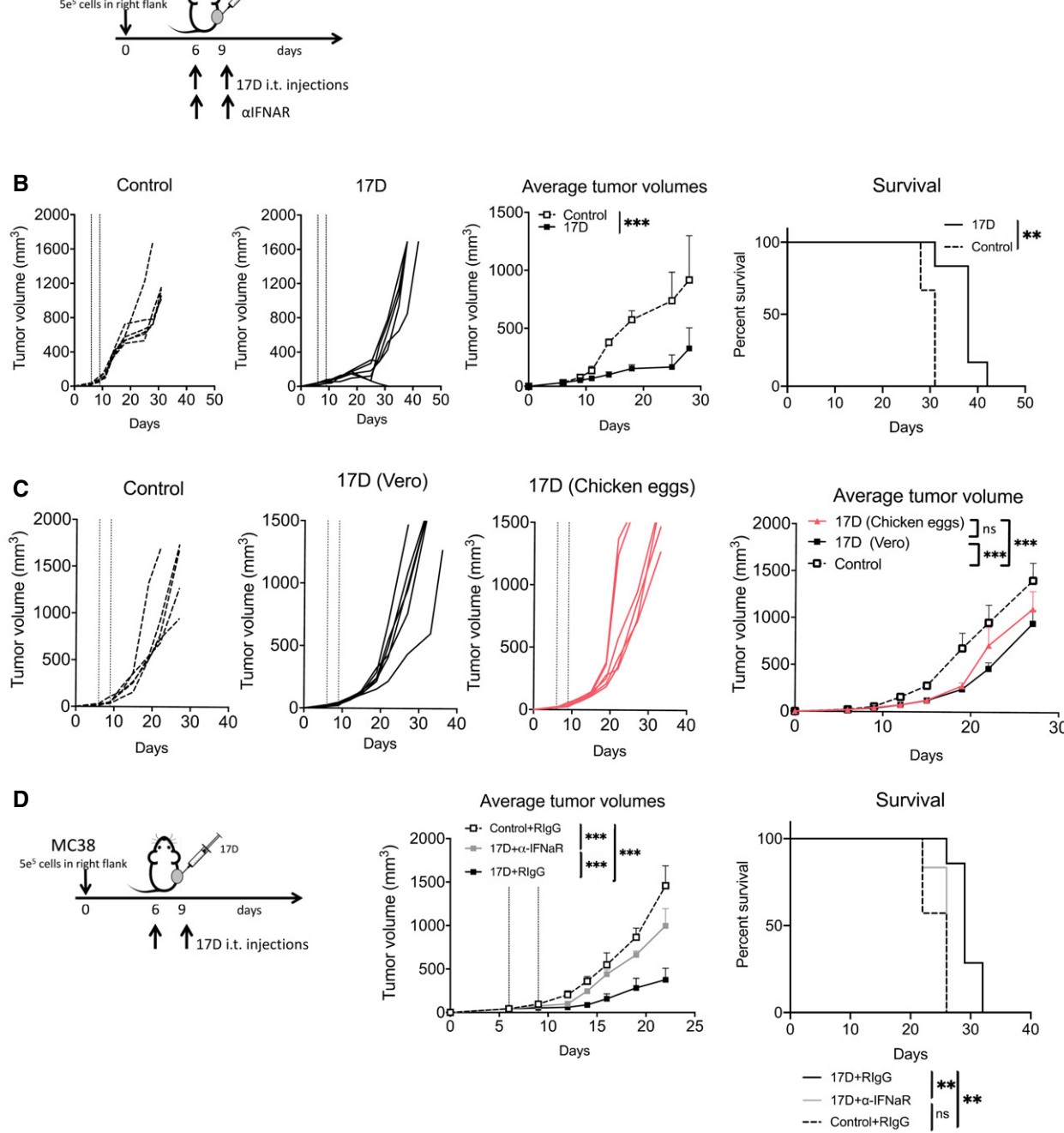

**Figure 8. 17D virus produced in chicken eggs exerts therapeutic effects on syngeneic transplantable mouse tumors and dependency of the antitumor effect on the type I IFN system.**

A   MC38 tumors were engrafted and treated with 17D as schematically represented. For *in vivo* IFNaR-1 blockade, anti-IFNaR-1 mAb was administered as indicated.

B   Individual tumor size follow-up upon intratumoral injections with 17D or vehicle as a control that are also shown as mean volume ± SD and as overall survival of the mice (n = 6 per group).

C   Individual tumor size follow-up upon intratumoral injections with 17D virus produced and purified from embryonated chicken eggs and vehicle as a control that are also shown as mean volume ± SD and as overall survival of the mice (17D+ IFNAR n = 5, 17D + RIgG and Control + RIgG n = 6).

D   Mean volume ± SD and overall survival of the mice treated with two intratumoral doses of 17D and with anti-IFNaR-1 blocking antibody (n = 5 for control group and n = 6 for each 17D-treated group).

Data information: Dashed lines indicate the days of intratumoral injection. Mean tumor volume growth over time was fitted using non-linear regression curve fit. Treatments were compared using the extra sum-of-squares *F*-test. Mantel–Cox test was used for survival analysis. Results are each from a single experiment performed. ***$P < 0.001$, **$P < 0.01$.

Source data are available online for this figure.

An important question is whether 17D is the most potent of the live viral vaccine strains currently in use in human preventive medicine. Comparisons with other vaccines will be undertaken in similar mouse models. In this regard, there are reported experiences with vaccinia virus that performs better when inactivated (Dai *et al*, 2017) and with measles virus (Msaouel *et al*, 2018). The role of preimmunization for these other agents needs to be addressed since increases in efficacy could be a general property of this type of local virotherapy (Ricca *et al*, 2018).

An important aspect described upon T-vec and NDV intratumoral treatment is that combinations with checkpoint inhibitors achieve additive or synergistic effects (Zamarin *et al*, 2014; Puzanov *et al*, 2016; Ribas *et al*, 2017; Ricca *et al*, 2018). In the case of 17D, clear synergy was observed with agonist anti-CD137 mAb against the directly treated tumor. Since many agents targeting this costimulatory molecule are being tested in the clinic (Chester *et al*, 2018), combinations with 17D with CD137 agonists should be feasible and advisable.

All considered, 17D is a candidate for an intratumoral dose escalation clinical trial in cancer patients who would be preimmunized with safe lower subcutaneous doses of 17D before intratumoral cycles of injections. Clinical trial design should follow the recently published expert recommendations for intratumoral virotherapy (Marabelle *et al*, 2018) in the context of biomarker finding research to monitor antiviral and antitumoral immune responses.

# Materials and Methods

### Mice

Six- to eight-week-old female C57BL/6 were purchased from Envigo (www.envigo.com). *Batf3*$^{-/-}$ and WT cousin colonies (Sanchez-Paulete *et al*, 2016) and TCR transgenic OT-1 CD45.1 mouse strain, all in C57BL/6 background, were bred in CIMA in specific pathogen-free conditions. WT and *Batf3*$^{-/-}$ from the heterozygotes and these colonies were used in subsequent experiments. All animal procedures were approved and conducted under institutional ethics committee guidelines (study number 040-16) with compliance with national, institutional, and EU guidelines. Mice were housed under a 12-h light/12-h dark cycle and had continuous access to food and water.

### Cell lines

Vero monkey kidney epithelial cells (ATCC CCL-81), mouse CT26 colon carcinoma, B16-F10 mouse melanoma cells, TC-1, mouse lung epithelial cells, and 4T1 mouse breast carcinoma were purchased from American Type Culture Collection (ATCC). B16-OVA melanoma cells and MC38 colon carcinoma cells were a kind gift from Dr. Lieping Chen (Yale University, New Haven, CT) and Dr. Karl E. Hellström (University of Washington, Seattle, WA), respectively. Human colon cancer cell lines HT-29 and HCT 116 were purchased from the ATCC. RCC10 human renal cell carcinoma was kindly provided by Dr Luis del Peso (CSIC-UAM, Madrid, Spain). BT-474 human breast cancer cell line was a kind gift from Dr López-Botet, IMIM, Barcelona, and UMBY and ICNI human melanoma were derived from primary surgical samples of metastatic lesions of patients at the Department of Dermatology, University Hospital Erlangen. Human non-transformed fibroblasts were cultured from normal tissue, a surgical specimen upon informed consent (F47550N) from our cell bank.

Vero cells were grown in DMEM with GlutaMAX (Gibco) containing 10% fetal bovine serum (FBS, Merck), 100 U/ml penicillin, and 100 μg/ml streptomycin (100 U/ml). Mouse tumor cells were cultured in RPMI 1640 with GlutaMAX (Gibco), 10% FBS, 100 U/ml penicillin and 100 μg/ml streptomycin (100 U/ml) (1% P/S), and 50 μM 2-mercaptoethanol (Gibco). Human HT-29, HCT 116, and RCC10 were cultured in the same medium without 2-mercaptoethanol. BT-474 was grown in DMEM/F12 (1:1) containing 2.5 mM glutamine (Invitrogen), 10% STF, and 1% P/S. UMBY and ICNI were grown in DMEM with GlutaMAX (Gibco) containing 10% FBS, 4 mM Gln, and 1% P/S. B16-OVA cells were supplemented with 400 μg/ml geneticin (Gibco).

Cell lines were routinely tested for mycoplasma contamination (MycoAlert Mycoplasma Detection Kit, Lonza). Cell lines were not authenticated for this project.

### 17D production, concentration, and quantification

To generate viral stocks, YF-17D virus (Stamaril, Sanofi Pasteur) was reconstituted as recommended by the manufacturer and propagated in Vero cells in DMEM with 2% STF and 1% P/S. Supernatants were collected at day 6 post-infection (when the cytopathic effect is > 80%) and clarified of debris by centrifugation at 2,000 *g* for 20 min at 4°C and stored at −80°C. Viral stocks were titrated as previously described (Fournier-Caruana *et al*, 2000). Briefly, serial tenfold dilutions of purified supernatants were inoculated to monolayers of Vero cells in 12-well plates and incubated 90 min at 37°C. Afterward, infectious supernatant dilutions were replaced by 2 ml overlay composed of Medium 199 (2×) (Thermofisher), 2% STF, 1% P/S, and 1.75% carboxymethyl cellulose (Merck). Six days post-infection, cells were washed with PBS, fixed 0.5% glutaraldehyde (Merck), and stained with 0.1% crystal violet dye (Panreac). The number of plaques for each dilution was then counted to estimate the plaque-forming units per ml (pfu/ml) of supernatant viral stocks.

To produce 17D for intratumoral injection, Vero monolayers were infected with 17D viral stocks at a multiplicity of injection (MOI) of 0.01–0.001 in DMEM 2% FBS. Infectious supernatants were harvested at day 6, centrifuged at 2,000 *g* for 20 min at 4°C, and subsequently ultracentrifuged at 30,500 *g* for 90 min at 4°C in 20% sucrose cushion. Virus was resuspended in TN buffer (Tris–HCl 50 mM pH 7.4, NaCl 100 mM). Single-use aliquots were stored at −80°C for *in vivo* experiments.

### *In vitro* sensitivity assay to 17D

Mouse and human tumor cell lines seeded into 24-well plates were infected at different MOIs for 90 min and were then grown in their corresponding culture medium. For mouse cell lines, the 6-day incubation was performed in low-serum (2% STF)-containing media. Human cells were cultured in 10% STF as several human cell lines stopped growing or detached at low STF conditions. Afterward, cells were washed, fixed, and stained with crystal violet. Crystal violet was dissolved in 10% acetic acid. Quantification was performed by

reading the OD values of the crystal violet-acetic acid solution in a micro-plate reader at 595 nm. The % of viability calculated is relative to non-17D-infected cells (100%). The Vero cell line was included as reference with both mouse and human cell lines. Infection at each MOI was done in three technical replicates per assay, and the assay was performed at least in two independent biological replicates per cell line, with different 17D batches.

### Intratumoral administration and *in vivo* efficacy experiments

MC38 colon carcinoma and B16-OVA melanoma were injected subcutaneously ($5 \times 10^5$) into the right flank of 7- to 10-week-old female C57BL/6 mice on day 0. Tumors were measured twice per week with calipers and the volume calculated (length $\times$ width$^2$/2). When tumors reached a mean volume of 125 mm$^3$ (on day 7 or 8 post-tumor inoculation), mice were randomized into different groups of treatment according to the experiment. 17D ($4 \times 10^6$ pfu in saline solution up to 50 µl of final volume) was administered by intratumoral injection twice per week for 2 weeks (four doses). The control group received intratumoral injections of 50 µl of identical volume of TN buffer (17D vehicle) in saline. Tumors were measured twice per week until the tumor volume reached the maximum allowed size or the animals died. Injections of 17D (either produced in Vero cells or in chicken eggs) on day 6 post-MC38 inoculation were performed as described above.

To evaluate the systemic antitumor effects, $5 \times 10^5$ (injected/treated tumor) and $3 \times 10^5$ (distant/untreated tumor) MC38 cells were injected into each flank, respectively. For evaluation of intratumoral 17D in combination with systemic immunostimulatory monoclonal antibodies, identical intratumoral treatment as described for single tumor models was performed, and mice received concomitant intraperitoneal administration (100 µg/dose) of either InVivoPlus anti-PD1 (RMP1-14), InVivo anti-CD137 (3H3), or InVivoMAb RIgG from BioXCell.

For qRT–PCR and flow cytometry experiments, mice received two intratumoral administrations and were euthanized 48 h post-second intratumoral injection.

### RNA extraction and quantitative RT–PCR

Total RNA was isolated in two steps using TRIzol (Life technologies) and RNeasy Mini-Kit (Qiagen) purification, following the manufacturer's RNA cleanup protocol and reverse transcription with M-MLV reverse transcriptase (Invitrogen). Quantitative RT–PCR (qRT–PCR) was performed with iQ SYBR Green Supermix in a CFX real-time PCR detection system (Bio-Rad). The following primer pairs were used to detect 17D gene expression (5'→3'): 17DFw ATGGAT GACTGGAAGAATGG, 17DRev GCTCCCTTTCCCAAATAGG, H3Fw AAAGCCGCTCGCAAGAGTGCG, H3Rev ACTTGCCTCCTGCAAAGC AC. Expression data were normalized with levels of the histone H3 housekeeping gene and represented according to the formula $2^{\Delta Ct \, (Ct_{H3 \text{ or } RNU6} - Ct \text{ gene})}$, where $C_t$ corresponds to cycle number.

### Depletion experiments

200 µg/dose of anti-CD4 (GK1.5), anti-CD8 (2.43), anti-NK (PK136) from BioXCell, were injected intraperitoneally 1 day before therapy. For GR-1$^+$ cell depletion, mice were treated with two doses of with

250 µg prior to 17D treatment. Subsequently, 150 µg of each depleting antibody was administered concurrently with intratumoral 17D. Then, mice were injected weekly for depletion maintenance (100 µg/dose). RIgG isotype (BioXCell) was injected as a control. Cell depletion was validated in blood samples by flow cytometry analysis. 160 µg/dose of anti-IFNaR-1 (IFN alpha/beta receptor subunit 1αR, clone MAR1-5A3) from BioXCell was injected intraperitoneally alongside with intratumoral injection of 17D. Two additional administrations of the blocking antibody were given every 3 days. RIgG isotype (BioXCell) was injected as a control.

### Flow cytometry

Fresh tumors were excised, weighed, and digested with 400 Mandl units/ml collagenase (Roche) and 50 mg/ml DNase (Roche) mixture for 15 min at 37°C. Enzymatic digestion was stopped with 12 µl/ml EDTA d 0.5 M, pH 8 (Gibco). Tumors were mechanically disrupted and filtered through a 0.7-mm cell strainer (BD Biosciences). After hypotonic lysis, single-cell suspensions from tumors and lymph nodes were treated with FcR-Block (anti-CD16/32 clone 2.4G2, 1:100 dilution) and were stained with Zombie NIR (BioLegend) and the following fluorochrome-conjugated antibodies (BioLegend): anti-CD45-PECy7 (30-F11 clone, 1:1,000 dilution), anti-CD8-BV510 (53-6.7 clone, 1:400), anti-NK1.1-PECy5 (PK-136 clone, 1:100), anti-CD137-APC (17B5, 1:200), anti-PD-1-FITC (29F.1A12, 1:100), anti-CTLA4 PE (UC10-4B9, 1:100), anti-CD45.2-FITC (104). 1:200, anti-CD25-APC (PC61, 1:400), anti-CD4-BV421 (GK1.5, 1:400), anti-CD45.1-BV785 (A20), anti-CD45.2-PerCP/Cy5.5 (104), anti-CD8-BUV395 (53-6.7, 1:250), anti-CD4-BUV496 (GK1.5,1:200), anti-CD3-AF700 (SK7, 1:250), anti-FoxP3 PE (FJK-16 S 1:200), Syr HAMST PE (SHG-1), Rat IgG1 APC (R3-34), Rat IgG2a FITC (RTK2758), Armenian hamster-PE (RTK4530), and mIgG1 PE (MOPC-21). All the isotype controls were incubated at the same final concentration as their corresponding test antibody. All antibodies were purchased from BioLegend. Cell suspensions were incubated with antibodies for 10 min at 4°C except for surface CTLA4 staining, which was performed 30 min at RT. For intracellular FOXP3 staining, cells were fixed, permeabilized, and stained using the True-Nuclear™ Transcription Factor Buffer Set (BioLegend) according to the manufacturer's protocol. To calculate absolute cell counts, perfect count microspheres (Cytognos) were used as an internal standard according to the manufacturer's instructions. Samples were acquired in a FACSCanto II (BD Biosciences). FlowJo (Treestar) software was used for data analysis.

### 17D immunization experiments and neutralization assay

Vaccination with 17D virus was performed as previously described (Gaucher *et al*, 2008) with a subcutaneous injection of $10^5$ pfu in 300 µl of PBS (s.c.) at the base of the tail. After 14 days, MC38 cells were bilaterally inoculated as described above to develop bilateral MC38 tumors. Intratumoral administration of 17D or control vehicle was performed at day 21 post-immunization.

To verify the immunization, a neutralization assay was adapted from Ricca *et al* (2018). Briefly, sera from 17D-immunized and 17D-naïve mice were collected at day 21 post-immunization and incubated with 17D at $10^4$ Pfu/ml 1 h at 37°C. Afterward, Vero cells were infected with the titration protocol described above and

**The paper explained**

**Problem**

Virotherapy is a modality of cancer treatment originally proposed to act mostly by exerting cytopathic activity on tumors but which mainly functions as a result of eliciting antitumor immune responses. A number of viruses either wild-type isolates or gene-transformed variants are being preclinically and clinically developed. Repurposing a safe and potent viral vaccine such as the yellow fever vaccine strain 17D for intratumoral injections would provide advantages from the regulatory and developmental points of view.

**Results**

Intratumoral injections of the yellow fever vaccine 17D manufactured by Sanofi Pasteur (Stamaril) exert antitumor effects in immunocompetent mouse models that impact non-injected lesions. The mechanism of action is mediated by CD8 T lymphocytes that infiltrate the tumor tissue microenvironment. The efficacy of treatment can be enhanced upon combination with immunomodulatory anti-CD137 monoclonal antibodies. Importantly, preimmunization to this virus does not hamper its intratumoral efficacy which, on the contrary, is markedly increased.

**Impact**

Intratumoral injections of 17D vaccine are to be clinically tested in combination with other immunotherapies for advanced cancer patients who can be preimmunized as a safety feature.

plaques were counted after 6 days. 17D containing serum from each mouse was used to infect two wells. Non-serum-incubated 17D was used as a control.

## Adoptive transfer of lymphocytes

To adoptively transfer of lymphocytes from 17D-immunized mice to naïve mice, 17D-immunized mice were euthanized 21–30 days after 17D immunization. $CD8^+$ and $CD4^+$ T cells were purified from spleens and lymph nodes with CD8 T-cell isolation kit mouse and CD4 T-cell isolation kit mouse (Miltenyi), respectively, following manufacturer's protocol. Purified T cells were intravenously injected ($10^7$ CD4 and/or CD8 per mouse) into mice bearing bilateral MC38 tumors, and 24 h later, they were intratumorally treated with 17D following the same treatment schedule as previously detailed.

To evaluate the differences between preimmunized and naïve lymphocytes, C57BL/6 mice were immunized with 17D, inoculated with B16OVA tumors after 14 days, and intravenously injected with OT-1 CD45.1 on day 20 ($8 \times 10^6$ per mouse). Afterward, mice received two intratumoral injections of 17D on days 21 (day 7 post-tumor inoculation) and 24 and euthanized 2 days after the second 17D dose.

## 17D UV inactivation

17D was inactivated by UV irradiation with three cycles of 15 min at 5 cm with a germicidal lamp spaced with three cycles of hood UV exposure. Virus inactivation was verified by titration on Vero.

## Virus growth in embryonated eggs

The production of 17D batches in embryonated chicken eggs was achieved via 17D inoculation in the yolk sac (Venter, 2014). After 6 days, the eggs were then placed at 4°C for 10 h before harvest. Evidence of the virus growth was confirmed by the presence of massive abnormalities in the embryos and the egg architecture as compared to a mock-inoculated egg. The virus was harvested from each egg and collected in individual falcon tubes. Tubes were centrifuged at 4,200 $g$ for 30 min, aliquoted, and quick frozen in dry ice. Virus was titrated by the Reed–Muench method in Vero cells 6 days after inoculation.

## Statistical analysis

Statistical analyses were performed using Prism software (GraphPad Software, Inc.). A two-tailed Student's $t$-test or Mann–Whitney tests were used to analyze statistical differences between two groups. Kruskal–Wallis test and post hoc comparisons were used to analyze the statistical differences between three or more groups. The Mantel–Cox test was used for survival analysis. For tumor growth data analyses, mean volumes of tumors over time were fitted using the formula $y = A \times e^{(t–t_0)}/(1 + e(t–t_0)/B)$, where $t$ represents time, A the maximum size reached by the tumor and B its growth rate. Treatments were compared using the extra sum-of-squares $F$-test. Values of $P < 0.05$ (*), $P < 0.01$ (**), and $P < 0.001$ (***) were considered significant. Exact $P$ values are shown in Appendix Table S1.

**Expanded View** for this article is available online.

## Acknowledgements

We thank MC Pinto for secretarial assistance. Drs. David Sancho, Juan J Lasarte, Miguel F. Sanmamed, and Jose L Perez Gracia are acknowledged for helpful discussions. This study was financially supported by the Spanish Ministry of Economy and Competitiveness (MINECO SAF2014-52361-R and FEDER/MICIU-AEI/SAF2017–83267-C2–1-R), Cancer Research Institute (CRI), Asociación Española contra el cancer (AECC), and Fundación BBVA.

## Author contributions

MAA and CM performed the *in vitro* experiments. MAA, CM, AA, SG, IR, and LC performed the *in vivo* and flow cytometry experiments. SR-R and EN-V performed the viral production in chicken eggs. MAA, CM, IE, AT, MA, LC, and ARS-P acquired the data. MAA, AT, IE, PB, and IM analyzed and interpreted the data. MAA, AT, MA, PB, and IM supervised the biological studies. MAA, PB, and IM designed the studies. MAA and IM drafted the work. MAA, CM, AT, AA, SG, ARS-P, IE, MAA, PB, and IM revised the manuscript. IM supervised the entire study.

## Conflict of interest

IM reports receiving commercial research grants from BMS, Alligator, and ROCHE and serves as a consultant/advisory board member for BMS, Merck-Serono, Roche-Genentech, Genmab, Incyte, Bioncotech, Tusk, Molecular partners F-STAR, Alligator, Bayer, Numab, Immunedesign, and AstraZeneca.

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
