## [Review Process File · EMBO Molecular Medicine]

Repurposing the Yellow Fever Vaccine for intratumoral Immunotherapy

M. Angela Aznar, Carmen Molina, Alvaro Teijeira, Inmaculada Rodriguez, Arantza Azpilikueta, Saray Garasa, Alfonso R. Sanchez-Paulete, Luna Cordeiro, Iñaki Etxeberria, Maite Alvarez, Sergio Rius-Rocabert, Estanislao Nistal-Villan, Pedro Berraondo, Ignacio Melero

Review timeline:	Submission date:	25 January 2019
	Editorial Decision:	25 February 2019
	Revision received:	3 September 2019
	Editorial Decision:	23 September 2019
	Revision received:	22 October 2019
	Accepted:	24 October 2019

Editor: Lise Roth

Transaction Report:

1st Editorial Decision

25 February 2019

Thank you for the submission of your manuscript to EMBO Molecular Medicine. We have now heard back from the 3 referees whom we asked to evaluate your manuscript.

As you will see from the reports below, the referees acknowledge the potential interest of the findings, the originality of the approach and the well-designed study, but they also have fundamental concerns (in particular regarding the statistical treatment of the results throughout the manuscript, and the limited clinical efficacy observed) that should be addressed in a major round of revision of the present manuscript.

Addressing the reviewers concerns in full will be necessary for further considering the manuscript in our journal. EMBO Molecular Medicine encourages a single round of revision only and therefore, acceptance or rejection of the manuscript will depend on the completeness of your responses included in the next, final version of the manuscript.

Please also contact us as soon as possible if similar work is published elsewhere. If other work is published, we may not be able to extend the revision period beyond three months.

I look forward to receiving your revised manuscript.

***** Reviewer's comments *****

Referee #1 (Comments on Novelty/Model System for Author):

The authors tested various preclinical models which reinforce the conclusion of the results

Referee #1 (Remarks for Author):

The authors repurpose a well known yellow fever vaccine, live 17 D as an oncolytic virus. Its advantage is that as a preventive vaccine, it has a good safety record. The authors demonstrated its efficacy in the injected lesions and contralateral tumors. When combined with anti-CD137 and more weakly with anti-PD-1, the clinical efficacy is enhanced. The experiments are well designed and controlled.

Some suggestions to strengthen the impact of this study :

- How do the authors explain the absence of neutralization of the virus, when mice were preimmunized and the mechanism explaining the enhanced clinical efficacy and CD8+T cell recruitment in the tumor.
- What about the specificity of 17 D infection towards normal human cells and the risk of toxicity.
- Fig 8 C : The effect of CD8 on contralateral lesions is modest when individual mice is shown compared to the average tumor volumes. It could be explained by the difference in the scale of monitoring (25-30 days in individual mice vs 20 days for the average). The scale for the average should be aligned to that of the individual mice (25-30 days)
- How do the authors explain that in the Fig 8B, the injection of 17D without CD8 transfer seems not efficient, while in Fig 2A, 17D alone leads to tumor regression.
- It could be added in the text that in some figures (Fig 6), results are discordant, when they were normalized by gram of tumors or total CD45.
- Fig 8 legends line 7 : The word tumor is repeated two times (It has to be deleted)

Referee #2 (Remarks for Author):

Authors propose the use of a vaccine against yellow fever virus (the attenuated yellow fever virus named 17D) as an oncolytic agent. This is a very original and potentially useful proposal. Experiments demonstrate the efficacy of intratumoral repeated administrations, especially when combined with immune stimulatory antibodies. A role for CD8 cells is also demonstrated. Interestingly, preimmunization with the vaccine enhances this efficacy.

Issues:

While in figures 2, 3 and 4 the 17D-treated groups had some efficacy, but in figure 5 the 17D-RIgG groups had no efficacy. In figures 7C and 8 the virus has no efficacy in naïve mice as well. Please explain these contradictions.

In general terms, the tumor growth control with the virus alone seems barely effective or ineffective, even with multiple intratumoral injections. As the relevance of the paper is proportional to this efficacy, it would be nice to see more single-agent efficacy. It would be nice to know the efforts done to improve this single-agent efficacy (higher virus doses, systemic administration of the virus...) or to understand why no more efficacy is achieved (quantify intratumoral replication, intratumoral barriers or anti-viral immunity).

For a practical application purpose it would be important to know if the available vaccine product (produced in eggs and containing different chicken products important for the vaccine efficacy) reproduces the results observed. Authors seem to use the vaccine product to propagate the D17 virus in Vero cells and later purify it in sucrose gradients and resuspend it in a buffer.

Minor points

Fig 2 in text is labeled as A,B,C, and D but only A and B in the real figure.

Given the large standard deviation of size in figure 3C (contralateral tumor growth) the statistical difference between groups seems unlikely. Please confirm.

Referee #3 (Comments on Novelty/Model System for Author):

1. In principle, the study is elaborate and conceptionally sound. However, the observed effects on tumor growth by the vaccine, which is repeatedly applied, are to some degree disappointing. There

also seem to be some flaws regarding the statistical analysis and calculated significance of the observed effects. Together, it is difficult to estimate their relevance and to interpret the mechanistical conclusions.

2. In total, novelty is medium: The use of the yellow fever vaccine as an 'oncolytic virus analogue' is novel and a clever idea. The use of an inflammatory agent to prepare tumors for checkpoint therapies is conceptionally of moderate novelty.

3. Regarding the medical impact, this already approved agent could be immediately included in clinical trials. On the other hand, synergy with the PD-1 antagonist is rather low, even with 4 applications of the vaccine i.t.. Combination with CD137 agonist is promising, but these agents are not that far in clinical development and have been associated with some toxicity.

Referee #3 (Remarks for Author):

In this study by Angela Aznar and colleagues, the authors investigated the use of intratumoral application of an approved live (attenuated) yellow fever vaccine to immunoactivate tumors for systemic checkpoint immunotherapy. The authors showed that the 17D vaccine is infectious and lytic in various human and murine cells, and delayed tumor growth in two immunocompetent models (MC38/B16-Ova). In a two-side MC38 model of lateral/injected and contralateral/noninjected tumors they found after vaccine injection an abscopal effect (no intratumoral detection of virus) in the noninjected, contralateral tumor. Depletion experiments were performed to demonstrate that these effects were mostly CD8 T cell dependent. The authors then evaluated the use of vaccine-dependent immunoactivation on concomitant checkpoint modulation with anti PD-1 or a CD137 agonist. Whereas PD-1 inhibition had rather modest effects, the CD137 agonist dramatically reduced tumor growth. This synergy seemed to be also detectable in the contralateral tumor, but to a much lesser degree. The authors also showed the modulatory effect of the vaccine on the immune cell contexture in the tumors finding a reduction of Tregs, an increase of CD8 with reduced exhaustion markers, and a decrease in NK which was interestingly inverted in the contralateral tumor. Unexpectedly, preimmunization before tumor establishment allowed for tumor growth inhibition by the intratumoral vaccine. Finally, by using lymphocyte transfer from immunized animals to tumor-bearing mice prior to tumor treatment, the authors wanted to show that the growth retardation by preimmunization was due to CD8 T cells.

In general, the topic of the study is interesting. How to convert immunologically 'cold' tumors into 'hot' ones to achieve responsiveness to checkpoint modulators is the central aim of current tumor immunotherapy including the ongoing clinical trials with oncolytic viruses (such as T-Vec plus Pembrolizumab). Since a live vaccine, directly applied to a tumor, is very similar to an oncolytic virus (see also Fig. 1 of the study), the basic concept is not fully novel. Nevertheless, to circumvent all efforts and regulatory issues associated with oncolytic virus development by using an already approved live vaccine, is a smart idea that could speed up the process. However, many of the shown effects are not fully convincing and there seem to be flaws in statistical evaluation making it difficult to rate the importance of the findings. In case of clear effects, such as the coapplication of the vaccine and a CD137 agonist on the treated tumor, there are already studies out describing the synergy using vaccinia viruses and the CD137 agonistic antibody (paper by John LB et al, Cancer Research 2012, p1651).

Major points:

An important point is the frequent vaccine application into the tumor (though effective oncolysis has been shown in Fig. 1) which may affect both tumors in many ways. It would be interesting to have an experiment with one or max. two vaccine applications plus the checkpoint modulators to estimate the real potency of the vaccine as a 'kick starter' for immunotherapy.

Fig. 2A: It seems that some of the treated mice in both groups dropped off earlier (due to tumor growth), but statistical evaluation is shown until days 30-40 which is inappropriate. This should be clarified e.g. by showing the curves only for the period with all participating individuals (as has been done in other figures). In the 17D group there seems to be one complete regression at day 21 or 22 - this did not lead to long-term survival? Please also check Fig. 2B. It should be clarified for which time points statistical significance has been achieved.

Fig. 3: Having in mind that only 6 individuals have been used, and looking at these individual tumor growth and the error bars in fig. C, I cannot see a significant difference in the contralateral tumor. Calculation should be re-checked. Is statistical significance maybe only valid for day 13? This time point is amidst the vaccine application regimen and could be also due to systemic interferon-

mediated effects rather than elicited T cell responses. At least in the equivalent groups in Fig. 4 the difference between Ctrl and 17D in the contralateral tumor seems to be more convincing.

Fig. 4/5: A survival analysis should be provided.

Fig 5: I can only see a relevant synergy of vaccine and aCD137 in the treated tumor.

Fig 7: The authors must use total tumor volume as in the previous figures and should not switch to 'relative to initial'. This way to illustrate tumor growth makes it difficult to estimate the relevance of the effects and to compare with tumor growth in the previous figures. Also, a group Control Naïve must be included here, as has been performed in the previous experiments.

Minor concerns

In the introduction, the authors mention 'unenestic' effects as an established term of describing effects on distant metastasis after treating a particular lesion. I do not know this term, a google search also did not help. My suggestion for a well established term would be 'abscopal' effect. In this context, the citation of the paper by Senzer et al, 2009, mentioning the elimination of distant melanoma lesions after intratumoral T-Vec injection would be a better fit than the used self-citations.

Sometimes, terms are not consistently used: The Isotype antibody is sometimes RIgG, sometimes RatIgG; Veh and Control in the legend of 5B/C. Should be standardized.

1st Revision - authors' response

3 September 2019

***** Reviewer's comments *****

We thank the reviewers for the constructive feedback that allowed us to improve the quality of our manuscript. We have thoroughly addressed reviewers' comments and herein we provide a point-by-point reply enclosed to a new and revised version in which two additional figures and an additional figure panels have been added or accordingly modified. The manuscript now has 10 figures and 5 supplementary files (EV files). Five additional literature references have been added concerning the experiments performed to address reviewers' comments (to a total of 62 references) and two additional authors who were crucial for 17D production in chicken eggs have been included. All authors are aware of these changes and agree to resubmission.

Our point-by-point responses to the comments are appended below. We hope that the questions and concerns have been fully addressed.

Referee #1 (Comments on Novelty/Model System for Author):

The authors tested various preclinical models, which reinforce the conclusion of the results
Referee #1 (Remarks for Author):

The authors repurpose a well-known yellow fever vaccine, live 17 D as an oncolytic virus. Its advantage is that as a preventive vaccine, it has a good safety record.

The authors demonstrated its efficacy in the injected lesions and contralateral tumors. When combined with anti-CD137 and more weakly with anti-PD-1, the clinical efficacy is enhanced. The experiments are well designed and controlled.

Some suggestions to strengthen the impact of this study:

- How do the authors explain the absence of neutralization of the virus, when mice were preimmunized and the mechanism explaining the enhanced clinical efficacy and CD8+T cell recruitment in the tumor.

Response: Intratumor administration of oncolytic viruses has been previously shown to be an efficacious way of bypassing the systemic anti-viral immunity. For instance, intratumorally injected adenoviruses were not neutralized by systemic antibodies in a subcutaneous pancreas ductal adenocarcinoma syrian hamster model, thereby allowing repeated administration of

the virus in the tumor bed {Li, 2017}. In this regard, Bassi and colleagues have shown that although anti-17D antibodies were present in the blood, 17D neutralization and elimination is not instantaneous, and 17D virus can be detected 3 days and even 5 days after 17D inoculation.

In fact, the only approved virotherapy so far is being administered intratumorally after previous seroconversion of the patients and has been shown that this doesn't impede tumor responses {Ribas, 2017}.

Work from Dr. Jedd Wolchok laboratory has established that in the case of intratumoral injections of New castle disease virus (NDV) to mouse tumors, preimmunization of the mice leads to better efficacy {Ricca et al, 2017}.

To evaluate the effects that 17D preimmunization may trigger in Tumor-infiltrating T lymphocytes after intratumoral 17D injection, we adoptively transferred 8×10^6 congenic CD8⁺ OT-1 CD45.1 to B16OVA-tumor bearing mice bilaterally which were previously preimmunized with 17D (Fig EV5 A). 48h after the second intratumoral injection of 17D to only one of the tumors, endogenous (CD45.2+) and transferred (tumor-specific OT-1 CD45.1+) T-cell infiltrates were assessed (Fig EV5 B).

The most striking feature is that the directly treated tumors in the preimmunized mice were much more profusely infiltrated by activated CD4 and to some extent endogenous CD8 T cells. In this regard, our results are concordant with those by Ricca et al {Ricca et al, 2017} using NDV.

In contralateral tumors, we observed a moderate (but not statistically significant) increase of endogenous CD45.2 CD8+ T cells per gram of tumor and a significant increase in frequencies of PD-1-expressing CD45.2+ CD8+ T cells (Fig EV 5C, upper panels). Interestingly, such changes in tumor-infiltrating lymphocytes were not occurring at this time-point in the transferred T cell subset (Fig EV 5C, lower panels). Moreover, intratumor CD8+ T cells of both endogenous and transferred subsets expressed PD-1 at lower levels (measured as Median Fluorescence Intensity) than the corresponding controls (Fig EV 5 C and D). Such PD-1 expression probably indicates a differential level of activation in these lymphocytes, rather than exhaustion phenotype (Ahn, Araki et al., 2018, Araki, Youngblood et al., 2013) since higher levels of expression are detected in chronic infection and exhaustion and represent terminally exhausted lymphocyte subsets (Blackburn, Shin et al., 2008, Wherry & Kurachi, 2015).

Taken together this results exposure to the viral antigens in the treated tumor of preimmunized mice seems to trigger a more intense and functional T-cell infiltrate in which Thelper cells probably play a substantial role. In this line, inflammation of the treated tumor probably helps at shaping a stronger antitumor response that to some extent also controls the progression of contralateral tumors

Action: New data are shown as EV5 of the revised version and we include new Material and Methods and new Results/discussion in the manuscript to report these findings.

- What about the specificity of 17D infection towards normal human cells and the risk of toxicity.

Response: To address this important question, we have performed the experiments of Fig 1 C in non-transformed human fibroblasts and compared the infection susceptibility to human tumor cell lines HCT-116 and ARST1. As shown panel C of Figure 1, non-transformed human cells were not sensitive to 17D infection at the same MOIs at which tumor cells were susceptible.

Action: new data are presented as Fig 1C, and new text has been included in Material and Methods and in Results section of the revised version to deal with these changes.

- Fig 8 C: The effect of CD8 on contralateral lesions is modest when individual mice is shown compared to the average tumor volumes. It could be explained by the difference in the scale of monitoring (25-30 days in individual mice vs 20 days for the average). The scale for the average should be aligned to that of the individual mice (25-30 days)

Response: Average tumor volumes are shown until control mice are dead (day 18). While mice are dying or being sacrificed because of the tumor growth, the average is recalculated with the tumor volume of living mice. Showing the average tumor volumes after some of the animals in the group are dead may lead to confusion and for this reason we represent

individual tumor follow-up in figures 9C and 9D. We now show a version of Fig 8 (Fig 1 only for reviewer, see figures appended below), including the number of mice alive in each time point to avoid confusion. The survival of these groups has been also included as an additional panel B.

Action: a version of Fig 8, renumbered now as Fig 9, only for Reviewers has been included at the end of this document.

- How do the authors explain that in the Fig 8B, the injection of 17D without CD8 transfer seems not efficient, while in Fig 2A, 17D alone leads to tumor regression.

Response: Although both experiments are performed in MC38 mouse models, the experiment of figure 2A shows mice bearing one tumor per mouse, while in Fig 9 (former Fig 8) each mouse are bearing two tumors, one of which was left uninjected. Following the reviewer comment, data has been thoroughly revised and we identified an error in some tumor volumes of the treated tumors. These data has been corrected and the figure has been updated. In addition, we have included the survival of mice in a new panel as Fig 9 B. We thank the Reviewer for this observation that allowed us to correct a mistake. Inter-experiment variability observed in antitumor responses to intratumoral 17D may be explained by the fact that in the present work we have produced 17D by ultracentrifugation and different 17D batches are being used along this experimental work. Although all conditions in 17D production are controlled, we cannot rule out that differences between batches may influence the results in the control of contralateral tumor, as this effects is mild (Fig 3a).

Action: We have corrected and updated Fig 8, and included an additional panel (Fig 9B) showing the survival among the different treatment groups.

- It could be added in the text that in some figures (Fig 6), results are discordant, when they were normalized by gram of tumors or total CD45.

Response: This discordance can be explained by the fact by the time of sacrifice controls and 17D treated tumors diverge in their tumor sizes, as 17D has been already administered twice, thus, for a given frequency of CD8 T cells in CD45 population, a smaller tumor size reflect a higher CD8+/gram tumor-tissue density.

- Fig 8 legends line 7: The word tumor is repeated two times (It has to be deleted)

Response: we thank Reviewer for indicating this. We have deleted this duplication. Action: duplicated word has been deleted from the manuscript.

Referee #2 (Remarks for Author):

Authors propose the use of a vaccine against yellow fever virus (the attenuated yellow fever virus named 17D) as an oncolytic agent. This is a very original and potentially useful proposal. Experiments demonstrate the efficacy of intratumoral repeated administrations, especially when combined with immune stimulatory antibodies. A role for CD8 cells is also demonstrated. Interestingly, preimmunization with the vaccine enhances this efficacy.

Issues:

While in figures 2, 3 and 4 the 17D-treated groups had some efficacy, but in figure 5 the 17D-RIgG groups had no efficacy. In figures 7C and 8 the virus has no efficacy in naïve mice as well. Please explain these contradictions.

These inconsistencies can be in part explained by the fact that we fixed the tumor size as a starting point of the intratumoral therapy. In our hands, and following our experimental protocol set-up in our lab, MC38 tumors reach 125 mm³ around day 7 or 8. Upon a careful

revision of our data sets, we have identified the experiments of figures 5, 7 and 8 as performed when tumors reached the fixed tumor volume on day 8. This delay on 17D intratumoral injection may account for these differences between figures and underlies the importance of the tumor microenvironment in 17D-mediated antitumor response over time. These results prompted us to evaluate 17D intratumoral injection in a different setting (on day +6 post-tumor injection, when tumors are already palpable) and with two doses of 4×10^6 PFU of 17D instead of four. This scheme leads to an intense therapeutic response in mice bearing a single MC38 tumor and these results are now included in Fig 10. In addition, we have repeated the experiment of pre-immunization shown in Fig 7 (renumbered as Fig 8) and the differences between 17D+RIgG and Control+RIgG groups are observed (Fig 8C).

Action: new experiments have been performed and resulted in a new Figure 8 and Figure 10A-C. New text has been included in Material and Methods and Result sections to include these changes.

In general terms, the tumor growth control with the virus alone seems barely effective or ineffective, even with multiple intratumoral injections. As the relevance of the paper is proportional to this efficacy, it would be nice to see more single-agent efficacy. It would be nice to know the efforts done to improve this single-agent efficacy (higher virus doses, systemic administration of the virus...) or to understand why no more efficacy is achieved (quantify intratumoral replication, intratumoral barriers or anti-viral immunity).

Response: As we observed that the differences observed in antitumor efficacy are in part dependent on the day of 17D therapy initiation, and with the objective of reducing the doses of 17D administered, we intratumorally injected 17D in MC38 tumor-bearing mice starting on day +6 and maintaining the dose of 17D constant. With this strategy, we observed a strong antitumor response in unilaterally MC38 implanted tumors. These results are dependent on type I IFN signaling, as blockade with an IFN α R blocking antibody almost completely abrogated such antitumor effects (Fig 10D). We thank the Reviewer for this feedback that has prompted us improving the quality of our work.

Action: new experiments have been performed and resulted in a new Fig 10 (Figures 10 B and C), and new text has been included in Material and Methods and Result section to deal with these changes.

For a practical application purpose it would be important to know if the available vaccine product (produced in eggs and containing different chicken products important for the vaccine efficacy) reproduces the results observed. Authors seem to use the vaccine product to propagate the D17 virus in Vero cells and later purify it in sucrose gradients and resuspend it in a buffer.

Response: To address this important question, we have produced 17D virus in embryonated eggs. Virus production in eggs presented some complications due to inefficient inoculation into the allantoic cavity of the egg. After several attempts, we managed to access the yolk sac by two means. One option was to brake and open the egg shell above the air sac by the use of a special boiled egg cracker (see images of figure 2, only for reviewers' inspection) and using an insulin syringe; the egg was sealed afterwards with sellotape. The second option was to open a small hole into the top to the egg above the air sac. In order to access the egg yolk or the amniotic sac, we used an insulin syringe with a 1 1/2 needle. The hole was sealed with wax. The inoculated eggs were incubated for 6 days. The eggs were then placed at 4 °C for 10 hours before harvest. Evidence of the virus growth could be observed inside by the massive abnormalities in the embryos and the egg architecture as compared to a mock inoculated egg (images 1C (control) and 1D). The virus was harvested from each egg and collected in individual falcon tubes. Tubes were centrifuged at 4200G for 30 minutes, aliquoted and quick frozen in dry ice. Virus was titrated by the Reed-Muench method in Vero cells 6 days after inoculation.

With this production, we evaluated the efficacy of 17D antitumor injection in mice bearing unilateral MC38 tumors implanted six days earlier. As is shown in Fig 10D, this treatment induced a strong antitumor response comparable to the virus produced in Vero cells.

Action: new Fig 10C has been included in the manuscript, and additional information has been added in Material and Methods and Results section to deal with these changes.

Minor points

Fig 2 in text is labeled as A,B,C, and D but only A and B in the real figure.

Response: We thank the reviewer for spotting this problem. The mislabeling has been corrected in Fig 2 and now all panels (A-D) are indicated.

Action: a new version of corrected Fig 2 has been included in the revised version of the manuscript.

Given the large standard deviation of size in figure 3C (contralateral tumor growth) the statistical difference between groups seems unlikely. Please confirm.

Response: Reviewer is right, we have revised the statistical analysis and confirmed that we have statistical significance until day 17 but it is lost at later time points (d21, see statistical analysis below). To clarify this, we have indicated statistical significance by asterisks only until day 17 in Figure 3B, and additional text explaining this fact has been included in Results section of the revised manuscript.

Action: Figure 3B has been corrected and additional text has been included in the results section of the manuscript.

Referee #3 (Comments on Novelty/Model System for Author):

1. In principle, the study is elaborate and conceptionally sound. However, the observed effects on tumor growth by the vaccine, which is repeatedly applied, are to some degree disappointing. There also seem to be some flaws regarding the statistical analysis and calculated significance of the observed effects. Together, it is difficult to estimate their relevance and to interpret the mechanistical conclusions.
2. In total, novelty is medium: The use of the yellow fever vaccine as an 'oncolytic virus analogue' is novel and a clever idea. The use of an inflammatory agent to prepare tumors for checkpoint therapies is conceptionally of moderate novelty.
3. Regarding the medical impact, this already approved agent could be immediately included in clinical trials. On the other hand, synergy with the PD-1 antagonist is rather low, even with 4 applications of the vaccine i.t. Combination with CD137 agonist is promising, but these agents are not that far in clinical development and have been associated with some toxicity.

Response: we agree with the overall appraisal summarized by the reviewer. However, even if not curative, the main virtue of the approach is its translational feasibility. With regard to CD137-based immunotherapy agents, we are aware of more than seven undergoing clinical trials or late preclinical research towards INDs. These are based on targeted bi-specifics and pro-body formats as well as 4-1BBL chimeric forms. We are in contact with one of these companies towards development of intratumoral injection of 17D in conjunction with this approaches.

Referee #3 (Remarks for Author):

In this study by Angela Aznar and colleagues, the authors investigated the use of intratumoral application of an approved live (attenuated) yellow fever vaccine to immunoactivate tumors for systemic checkpoint immunotherapy. The authors showed that the 17D vaccine is infectious and lytic in various human and murine cells, and delayed tumor growth in two immunocompetent models (MC38/B16-Ova). In a two-side MC38 model of lateral/injected and contralateral/noninjected tumors they found after vaccine injection an abscopal effect (no intratumoral detection of virus) in the noninjected, contralateral tumor. Depletion experiments were performed to demonstrate that these effects were mostly CD8 T cell dependent. The authors

then evaluated the use of vaccine-dependent immunoactivation on concomitant checkpoint modulation with anti PD-1 or a CD137 agonist. Whereas PD-1 inhibition had rather modest effects, the CD137 agonist dramatically reduced tumor growth. This synergy seemed to be also detectable in the contralateral tumor, but to a much lesser degree. The authors also showed the modulatory effect of the vaccine on the immune cell contexture in the tumors finding a reduction of Tregs, an increase of CD8 with reduced exhaustion markers, and a decrease in NK which was interestingly inverted in the contralateral tumor. Unexpectedly, preimmunization before tumor establishment allowed for tumor growth inhibition by the intratumoral vaccine. Finally, by using lymphocyte transfer from immunized animals to tumor-bearing mice prior to tumor treatment, the authors wanted to show that the growth retardation by preimmunization was due to CD8 T cells. In general, the topic of the study is interesting. How to convert immunologically 'cold' tumors into 'hot' ones to achieve responsiveness to checkpoint modulators is the central aim of current tumor immunotherapy including the ongoing clinical trials with oncolytic viruses (such as T-Vec plus Pembrolizumab). Since a live vaccine, directly applied to a tumor, is very similar to an oncolytic virus (see also Fig. 1 of the study), the basic concept is not fully novel. Nevertheless, to circumvent all efforts and regulatory issues associated with oncolytic virus development by using an already approved live vaccine, is a smart idea that could speed up the process. However, many of the shown effects are not fully convincing and there seem to be flaws in statistical evaluation making it difficult to rate the importance of the findings. In case of clear effects, such as the coapplication of the vaccine and a CD137 agonist on the treated tumor, there are already studies out describing the synergy using vaccinia viruses and the CD137 agonistic antibody (paper by John LB et al, Cancer Research 2012, p1651).

Major points:

An important point is the frequent vaccine application into the tumor (though effective oncolysis has been shown in Fig. 1) which may affect both tumors in many ways. It would be interesting to have an experiment with one or max. two vaccine applications plus the checkpoint modulators to estimate the real potency of the vaccine as a 'kick starter' for immunotherapy.

Response: To address this important question, we have treated mice bearing established MC38 tumors for 6 days with similar doses of the virus and observed remarkable therapeutic effects in these mice bearing unilateral tumors.

Action: New experiments are in figure 10B and figure 10D. These observation upon earlier treatment of the tumors are commented on in results and discussed.

Fig. 2A: It seems that some of the treated mice in both groups dropped off earlier (due to tumor growth), but statistical evaluation is shown until days 30-40 which is inappropriate. This should be clarified e.g. by showing the curves only for the period with all participating individuals (as has been done in other figures). In the 17D group there seems to be one complete regression at day 21 or 22 - this did not lead to long-term survival? Please also check Fig. 2B. It should be clarified for which time points statistical significance has been achieved.

Reviewer is right, and therefore we have corrected the figure and show average tumor volumes until first mouse death (corresponding to control group, day 18). Statistical analysis of tumor growth control is done by analyzing the differences in the growth curves along the whole experiment. For tumor growth data analyses, mean volumes of tumors over time were fitted using the formula $y = A \times e^{-(t-t_0)} / (1 + e^{-(t-t_0)/B})$, where t represents time. Therefore, this statistical method informs about the differences in the growth curve, not only in the differences in tumor sizes in particular data point. We have redone the analysis and confirm the same statistical significance at day 18.

On the other hand, the mouse that seems to regress reached 13.5 mm³ but eventually was found dead and was maintained in the study as no signs or disease different from the tumor were observed.

Action: Figure 2 has been corrected and new average tumor growth is shown in Fig 2B as well as overall survival.

Fig. 3: Having in mind that only 6 individuals have been used, and looking at these individual tumor growths and the error bars in fig. C, I cannot see a significant difference in the contralateral

tumor. Calculation should be re-checked. Is statistical significance maybe only valid for day 13? This time point is amidst the vaccine application regimen and could be also due to systemic interferon-mediated effects rather than elicited T cell responses. At least in the equivalent groups in Fig. 4 the difference between Ctrl and 17D in the contralateral tumor seems to be more convincing.

The reviewer is right, we have revised the statistical analysis and confirmed that we have statistical significance until day 17 but this is lost on later time points. To clarify this, we have indicated in the figure and in the results section that statistical significance is lost after day 17 in contralateral tumors.

Action: Figure 3B has been corrected and additional text has been included in the results section of the manuscript.

Fig. 4/5: A survival analysis should be provided.

We thank the Reviewer for this observation, and a survival analyses are now provided.

Action: survival has been included for experiments depicted in Fig 4 and 5 (shown in panel D in each figure).

Fig 5: I can only see a relevant synergy of vaccine and aCD137 in the treated tumor.

As Reviewer indicates, a major systemic effect is reached in treated tumors when 17D intratumoral injection is combined with anti-CD137 mAb. However, systemic anti-CD137 per se is strong enough to induce an important delay in contralateral tumor growth, and in this setting 17D did not add further efficacy against the contralateral tumor. This has been indicated in the text in Results section.

Action: new text in Results section has been included to address Reviewer's feedback.

Fig 7: The authors must use total tumor volume as in the previous figures and should not switch to 'relative to initial'. This way to illustrate tumor growth makes it difficult to estimate the relevance of the effects and to compare with tumor growth in the previous figures. Also, a group Control Naïve must be included here, as has been performed in the previous experiments.

Response: to better illustrate the tumor growth and maintain the logics of all the figures shown in the manuscript, we have replaced the previous data of Fig 7 by newly performed experiment for Fig 7 (now renumbered as Fig 8 the single experiment performed during the revision process including Control Naïve group as per suggested by the Reviewer).

Action: new experiment for is included as Fig 8 in the revised version of the manuscript.

Minor concerns

In the introduction, the authors mention 'unenestic' effects as an established term of describing effects on distant metastasis after treating a particular lesion. I do not know this term, a google search also did not help. My suggestion for a well established term would be 'abscopal' effect. In this context, the citation of the paper by Senzer et al, 2009, mentioning the elimination of distant melanoma lesions after intratumoral T-Vec injection would be a better fit than the used self-citations.

Action: as suggested by the Reviewer, we have added the term *abscopal*, which better designs the systemic effect induced in distant lesions upon local treatment.

Sometimes, terms are not consistently used: The Isotype antibody is sometimes RIgG, sometimes RatIgG; Veh and Control in the legend of 5B/C. Should be standardized.

We thank the reviewer for this observation, and we have corrected these terms in figures and figure legends of the revised version of the manuscript.

Action: we have replaced the terms RatIgG by RIgG in figures and in the figure legend of Fig 5.

FIGURES FOR REVIEWERS' INSPECTION

Figure 1

Figure 2

Figure 3

2nd Editorial Decision

23 September 2019

Thank you for the submission of your revised manuscript to EMBO Molecular Medicine. We have now received the referees' reports, and as you will see they are now supportive of publication of your work. I am therefore pleased to inform you that we will be able to accept your manuscript once minor editorial concerns are addressed.

I look forward to reading a new revised version of your manuscript as soon as possible.

***** Reviewer's comments *****

Referee #1 (Comments on Novelty/Model System for Author):

Originality of the repurposing of yellow fever vaccine
Various experimental models
Demonstration that the pre_immunization status does not preclude the efficacy of the oncolytic virus

Referee #1 (Remarks for Author):

The authors satisfactorily address my various concerns

Referee #2 (Remarks for Author):

None

Referee #3 (Comments on Novelty/Model System for Author):

1. The study is elaborate and conceptionally sound. Experiments have been performed with all necessary control groups.
2. The novelty is good since the use of the yellow fever vaccine as an oncolytic Virus analogue off-the-shelf is a smart idea. The use of an oncolytic or inflammatory agent to immunoactivate a tumor for checkpoint agonists or antagonists is an already established concept.
3. Medical Impact is high since already components can be used that are either approved or are in late clinical development. Particularly the synergy with anti-CD137 looks promising.
4. The MC38 model in Black6 is well established and suitable for this study.

Referee #3 (Remarks for Author):

My concerns have been appropriately addressed.

2nd Revision - authors' response

22 October 2019

Authors made the requested editorial changes.

Corresponding Author Name: Ignacio Melero Bermejo

Journal Submitted to: EMBO MM

Manuscript Number: EMM-2019-10375